# Taming Hyperparameter Sensitivity in Data Attribution: Practical Selection Without Costly Retraining

**Weiyi Wang**[1]    **Junwei Deng**[2]    **Yuzheng Hu**[2]    **Shiyuan Zhang**[2]    **Xirui Jiang**[1]
**Runting Zhang**[1]    **Han Zhao**[2]    **Jiaqi W. Ma**[2]

[1]University of Michigan Ann Arbor    [2]University of Illinois Urbana-Champaign

{wweiyi,xirui,runtingz}@umich.edu
{junweid2,yh46,sz54,hanzhao,jiaqima}@illinois.edu

## Abstract

Data attribution methods, which quantify the influence of individual training data points on a machine learning model, have gained increasing popularity in data-centric applications in modern AI. Despite a recent surge of new methods developed in this space, the impact of hyperparameter tuning in these methods remains under-explored. In this work, we present the first large-scale empirical study to understand the hyperparameter sensitivity of common data attribution methods. Our results show that most methods are indeed sensitive to certain key hyperparameters. However, unlike typical machine learning algorithms—whose hyperparameters can be tuned using computationally-cheap validation metrics—evaluating data attribution performance often requires retraining models on subsets of training data, making such metrics prohibitively costly for hyperparameter tuning. This poses a critical open challenge for the practical application of data attribution methods. To address this challenge, we advocate for better theoretical understandings of hyperparameter behavior to inform efficient tuning strategies. As a case study, we provide a theoretical analysis of the regularization term that is critical in many variants of influence function methods. Building on this analysis, we propose a lightweight procedure for selecting the regularization value without model retraining, and validate its effectiveness across a range of standard data attribution benchmarks. Overall, our study identifies a fundamental yet overlooked challenge in the practical application of data attribution, and highlights the importance of careful discussion on hyperparameter selection in future method development.

## 1 Introduction

Data attribution methods quantify the contribution of individual training samples to a machine learning model's predictions or overall performance [22, 9]. These methods have been successfully applied to a range of data-centric tasks in modern AI, including training data selection [37], mislabel detection [22], and copyright compensation [7]. Recently, this field has witnessed a surge of novel methods [11, 18, 30, 12, 29, 6, 3, 35], each promising improvements in efficiency, efficacy, or theoretical soundness.

Despite this rapid progress, a crucial practical aspect remains largely underexplored: the **hyperparameter sensitivity** of data attribution methods. Existing benchmark studies [19, 8] largely adopt the hyperparameter settings used in the original papers proposing the methods. While some of the original papers include limited ablation studies, few of them offer practical guidance for hyperparameter selection, and many methods include implicit tunable knobs that have received little scrutiny. Moreover, the optimal hyperparameter choices can vary across datasets and models.

39th Conference on Neural Information Processing Systems (NeurIPS 2025).

Hyperparameter tuning is particularly challenging in data attribution due to the high computation costs associated with common evaluation metrics. Standard evaluation metrics—such as Leave-One-Out (LOO) correlation [22] and Linear Datamodeling Score (LDS) [29]—require retraining models on subsets of the full training dataset for hundreds or even thousands of times. A conventional hyperparameter search under such evaluation metrics is prohibitively expensive in practice.

In this work, we present the first systematic, large-scale empirical study of hyperparameter sensitivity in data attribution. We benchmark a suite of popular data attribution methods across various datasets and models with extensive hyperparameter sweeps. Our results confirm that most methods are sensitive to certain key hyperparameters, and the optimal hyperparameter choices vary across experimental settings. Our analysis also reveals other interesting novel findings regarding the *interaction* of different hyperparameters. Our findings highlight hyperparameter tuning as a critical open challenge in the practical application of data attribution.

As a first step towards addressing this challenge, we conduct a case study on selecting the regularization term—a crucial hyperparamter in many influence function methods [22, 12]—without costly retraining. We first provide a theoretical analysis of how the regularization term affects the influence function performance in terms of LDS. Building on this analysis, we introduce a surrogate indicator that enables selection of the regularization value without model retraining. We validate the effectiveness of the proposed approach across multiple experimental settings, including several variants of influence functions on different datasets and models.

To summarize, our contributions are as follows:

- We identify hyperparameter tuning as a critical but largely overlooked challenge in the practical application of data attribution methods.

- We conduct the first comprehensive empirical study of hyperparameter sensitivity in data attribution, benchmarking a range of widely used methods across diverse settings, confirming the necessity of practical hyperparameter selection strategies.

- We provide a theoretical analysis of the regularization term in influence functions and propose an efficient, retraining-free selection procedure with strong empirical performance.

These results advance both the understanding and practical deployment of data attribution methods, encouraging a more systematic treatment of hyperparameters as integral components in future methodological development.

## 2 Background

In this section, we provide the necessary background to contextualize our empirical study of hyperparameter sensitivity in Section 3 and the subsequent case analysis in Section 4.

### 2.1 Data Attribution Methods and Their Hyperparameters

In this work, we focus on the methods included in a recent benchmark study [8], which consists of state-of-the-art efficient data attribution methods while omitting a family of game-theoretic methods such as Data Shapley [11, 18].[1] In the following, we introduce the notation, then review a set of most widely-used data attribution methods as well as their major hyperparameters examined in our study.

**Notation.** Consider a multi-class classification problem with dataset $S = \{(x_i, y_i)\}_{i=1}^n \subset \mathcal{Z}$, where $\mathcal{Z} = \mathcal{X} \times \mathcal{Y}$, inputs $x_i \in \mathcal{X} \subset \mathbb{R}^d$, and labels $y_i \in \mathcal{Y}$. We use a parameterized hypothesis function $h(\cdot, \theta) : \mathcal{X} \to \Delta_{|\mathcal{Y}|-1}$ with parameters $\theta \in \mathbb{R}^p$, which typically outputs class probabilities via a softmax layer. The cross-entropy loss for a single example $z = (x, y)$ is denoted by $L(z, \theta) = \ell(h(x, \theta), y)$. The empirical risk at $\theta$ over a subset $A \subset S$ of size $a$ is $R_A(\theta) := \frac{1}{a} \sum_{z \in A} L(z, \theta)$. For a given test example $z' = (x', y')$, following Park et al. [29], we define the model output function as $f(z', \theta) := \ln \frac{p(z', \theta)}{1 - p(z', \theta)}$, where $p(z', \theta) = \exp(-L(z', \theta))$ is the model's predicted probability for the correct class $y'$. We denote the parameter that minimizes $R_S$ as $\theta_S^*$.

---

[1] While these game-theoretic methods are theoretically principled, they are typically computationally expensive and cannot scale up to even moderately large models.

**Data attribution methods.** A data attribution method (or an *attributor*) is a function $\tau : \mathcal{Z} \times S \to \mathbb{R}$ that, for a given test example $z' \in \mathcal{Z}$, assigns $\tau(z', z_i)$ to each training point $z_i \in S$. A popular attributor is the Influence Function (IF) attributor, introduced by Koh and Liang [22], which computes the influence of a training example $z_i \in S$ on a test example $z' \in \mathcal{Z}$ as

$$\tau_{\text{IF}}(z', z_i) := -\nabla_\theta f(z', \theta_S^*)^\top H_S^{-1} \nabla_\theta L(z_i, \theta_S^*), \tag{1}$$

where $H_S := \nabla_\theta^2 R_S(\theta_S^*)$ is the Hessian of the empirical risk $R_S$. To bypass the prohibitive computation of $H_S$, they also propose using efficient approaches to approximate $H_S^{-1} \nabla_\theta L(z_i, \theta_S^*)$, including conjugate gradient (CG) [27] and stochastic estimation method LiSSA [1]. For non-convex models, $H_S$ is usually replaced with its *regularized* version $H_S + \lambda I_p$ in case of singularity [22], where $\lambda > 0$ is a regularization term and $I_p$ is the identity matrix of size $p$. To further handle non-convexity and reduce computational costs, Grosse et al. [12] draw from second-order optimization literature [26] and introduce the Generalized Gauss Newton matrix (GGN) and empirical Fisher Information Matrix (FIM) to approximate $H_S$, which are guaranteed to be positive semi-definite (or even positive definite when a small regularization $\lambda I_p$ is included), and only involve first-order differentiation. We formally define the Influence Function attributor with FIM (IFFIM) as

$$\tau_{\text{IFFIM},\lambda}(z', z_i) := -\nabla_\theta f(z', \theta_S^*)^\top (F_S + \lambda I_p)^{-1} \nabla_\theta L(z_i, \theta_S^*), \tag{2}$$

where $F_S$ is the empirical FIM defined as:

$$F_S := \frac{1}{n} \sum_{z_i \in S} [\nabla_\theta L(z_i, \theta_S^*) \nabla_\theta L(z_i, \theta_S^*)^\top] = \mathbb{E}_{z_i \sim S}[\nabla_\theta L(z_i, \theta_S^*) \nabla_\theta L(z_i, \theta_S^*)^\top]. \tag{3}$$

The TRAK [29] and LoGra [6] attributors adopt similar forms[2] while further introducing *gradient projections* to accelerate computation. In this case, we have the following modified IFFIM attributor, which takes a projection matrix $P \in \mathbb{R}^{p \times \tilde{p}}$ with projection dimension $\tilde{p}$ into consideration:

$$\tau_{\text{IFFIM},\lambda,P}(z', z_i) := -\nabla_\theta f(z', \theta_S^*)^\top P(P^\top F_S P + \lambda I_{\tilde{p}})^{-1} P^\top \nabla_\theta L(z_i, \theta_S^*). \tag{4}$$

Another popular method, TracIn [30], computes the influence of $z_i$ on $z'$ by tracing the entire optimization process based on stochastic gradient descent. Mathematically, it works as if substituting $H_S^{-1}$ with the identity matrix, while taking average across training model checkpoints weighted by the learning rate: $\tau_{\text{TracIn}}(z', z_i) := \sum_t \eta_t \nabla_\theta f(z', \theta_t)^\top \nabla_\theta L(z_i, \theta_t)$, where $\theta_t$ and $\eta_t$ are the parameters and learning rate at checkpoint $t$.

**Major hyperparameters.** We highlight some major hyperparameters that are shared across multiple data attribution methods mentioned above.

- Regularization term $\lambda$: the regularization term $\lambda$ for handling non-convexity is a common hyperparameter across many methods, including IF [22], TRAK [29], LoGra [6], etc.

- Projection dimension $\tilde{p}$: gradient projection is a common practice to reduce computational costs in data attribution methods [33, 6, 29], making the projection dimension $\tilde{p}$ another common hyperparameter.

- Training epoch: while, in principle, most data attribution methods rely on the optimal model parameters $\theta_S^*$ to calculate the attribution scores, one can also calculate the attribution scores using an earlier model checkpoint $\theta_t$ in practice. This makes the *training epoch* of the checkpoint used to calculate the attribution scores an implicit hyperparameter, whose impact is largely overlooked in the literature.[3]

## 2.2 Evaluation Metrics for Data Attribution

Evaluation metrics for data attribution can be categorized into application-agnostic and application-specific metrics.

The application-agnostic metrics include Leave-One-Out (LOO) correlation [22], Linear Datamodeling Score (LDS) [29], and Brittleness [16]. In general, these metrics measure the performance

---

[2]Appendix B.1 provides a detailed discussion on the relationship between TRAK and IFFIM.

[3]TracIn [30] explicitly uses multiple checkpoints as shown in the definition of $\tau_{\text{TracIn}}$ while TRAK [29] also explores the effect of aggregating over different checkpoints in an ablation study.

of data attribution methods by their ability to predict the model output when training on subsets of the training data. As a result, they typically require hundreds or even thousands of model retraining on subsets of the training dataset. While these metrics are widely used as principled evaluation criteria for novel data attribution methods in research setup, they are prohibitively expensive for hyperparameter tuning in practical applications.

The application-specific metrics rooted in individual downstream application scenarios. Two common scenarios include noisy label detection and data selection [22, 11]. While these metrics are typically less computationally demanding, they are restricted to specific applications and only provide a partial picture of the quality of data attribution. Hyperparameter tuning in data selection also relies on retraining the model on the selected data subset, which can be expensive for large-scale applications such as large language models [37].

In this work, we focus on LDS, which is one of the most widely used evaluation metrics in recent literature [29, 6, 8], for both the hyperparameter sensitivity study and the case study. Formally, LDS is defined as following.

**Definition 2.1** (Linear Datamodeling Score (LDS) [29]). *Given an attributor $\tau$ and a test example $z' \in \mathcal{Z}$, sample $s$ subsets $\overline{A} = \{A_1, \cdots, A_s\}$, where each $A_j \subset S$ is sampled uniformly at random with fixed size $a$. The* Linear Datamodeling Score (LDS) *of $\tau$ for $z'$ on $\overline{A}$ is defined as:*

$$c_s(\tau, z', \overline{A}) := \mathrm{SpearmanCorr}\left( \{f(z', \theta^*_{A_j})\}_{j \in [s]}, \big\{ \sum_{z \in A_j} \tau(z', z) \big\}_{j \in [s]} \right), \tag{5}$$

*where $\theta^*_{A_j}$ is the optimal model parameters learned on a subset $A_j$, the subset influence is modeled as additive in individual influences [15], and $\mathrm{SpearmanCorr}$ is the Spearman correlation [34].*

## 2.3 Related Work on Hyperparameter Sensitivity in Data Attribution

Finally, we review existing literature with dedicated discussions on the effect of hyperparameters in data attribution. Koh et al. [23] examine the change in attribution quality when the regularization strength $\lambda$ is tuned in IF settings. They empirically observe an enhancement of the correlation between the attributed influences and actual effects when $\lambda$ is increased, and provide an upper bound on the approximation error of IF which decreases with $\lambda$. Basu et al. [4] empirically study the effects of neural network architectures and weight-decay during model training on IF attribution quality. They reveal an increase in error when the network is deep or weight-decay is absent. Zhang and Zhang [38] utilize techniques from Neural Tangent Kernel [17] to provide a theoretical lower bound of the LOO counterfactual loss change for wide two layer ReLU networks with regularized mean-square loss. Klochkov and Liu [21] leverage the spectral properties of Hessian to guide selection of hyperparameters specific in LiSSA. However, to our knowledge, this work is the first systematic, large-scale study on the hyperparameter sensitivity in data attribution that points out the unique computational challenge for hyperparameter tuning in practice.

# 3 A Large-Scale Study of Hyperparameter Sensitivity in Data Attribution

## 3.1 Experimental Setup

**Data attribution methods.** We evaluate a range of data attribution methods, including IF and its variants (CG and LiSSA) [22], TracIn [30], TRAK [29], and LoGra [6], introduced in Section 2.1.

**Datasets and models.** We perform empirical evaluations on three standard benchmark settings: MNIST [25] with a multilayer perceptron (MLP), CIFAR-2 [24] with ResNet-9 [14], and Wiki-Text2 [28] with the GPT2 [31], which consists of both image and text data with various model sizes. We use the implementation from the `dattri` library [8] for the datasets and models.

**Experiment design.** We carry out the comprehensive study through multiple sub-experiments. For each sub-experiment, we analyze the sensitivity of one or two hyperparameters while controlling other hyperparameters to be fixed. We perform a grid search over the chosen hyperparameters within the search space. We calculate LDS for each experiment under 50 independently retrained models.

**Hyperparameter selection and search space.** We consider both the common hyperparameters introduced in Section 2.1 and some critical method-specific hyperparameters. For TRAK, we experiment with regularization, projection-dimension, and training-epoch. For TracIn, we search for projection-dimension, normalization, and checkpoint-selection. For IF, we analyze regularization and training-epoch, as well as max-iteration for the CG variant, and scaling and recursion-depth for the LiSSA variant. For LoGra, we search for regularization, projection-dimension, and training-epoch. We design the search space for each hyperparameter around its default value proposed by the original papers. The detailed definitions and the search space of the hyperparameters are stated in Appendix A.1.

## 3.2 Experimental Results

We summarize our findings and insights from the experimental results (Figure 1)[4] as follows:

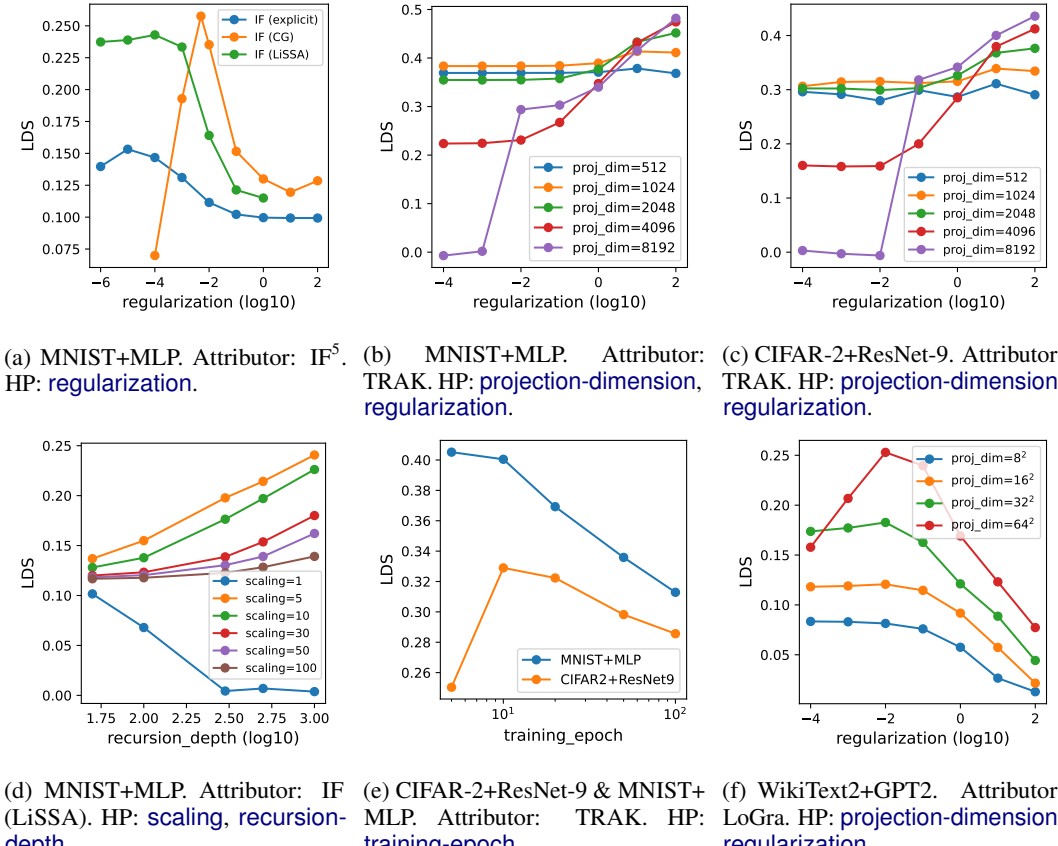

(a) MNIST+MLP. Attributor: IF[5]. HP: regularization.

(b) MNIST+MLP. Attributor: TRAK. HP: projection-dimension, regularization.

(c) CIFAR-2+ResNet-9. Attributor: TRAK. HP: projection-dimension, regularization.

(d) MNIST+MLP. Attributor: IF (LiSSA). HP: scaling, recursion-depth.

(e) CIFAR-2+ResNet-9 & MNIST+MLP. Attributor: TRAK. HP: training-epoch.

(f) WikiText2+GPT2. Attributor: LoGra. HP: projection-dimension, regularization.

Figure 1: Part of the experimental results of hyperparameter sensitivity analysis.

- **Most attribution methods are sensitive to certain hyperparameters.** Small changes in hyperparameters can lead to large variations in LDS performance, highlighting the sensitivity of most methods to hyperparameter choices. For example, Figure 1a shows that increasing regularization from 1e-4 to 1e-2 for IF (CG) leads to a 3× increase in LDS.
- **The best hyperparameter choices vary across datasets and models.** A configuration that performs well for one setting (dataset and model) may not transfer to another setting, reinforcing the need for hyperparameter tuning for each specific task. For example, Figure 1e shows that

---

[4]Some additional results are shown in Section A.2.

[5]IF (explicit) refers to the original definition of IF that explicitly inverts the Hessian. Only the last layer parameters are used for IF (explicit) due to the memory limit.

setting training-epoch as 5 on MNIST+MLP is the best hyperparameter choice while leading to poor performance on CIFAR-2+ResNet-9.

- **The impact of hyperparameters can be entangled.** We find that the choice of one hyperparameter can affect the sensitivity of LDS to others. As an example, the LDS of both TRAK and LoGra becomes more sensitive to regularization for higher projection-dimension, as depicted in Figure 1b, Figure 1c, and Figure 1f. Intuitively, higher projection-dimension implies that the Hessian or empirical FIM is closer to being singular, which leads to more sensitive attribution scores. Interestingly, the entanglement of regularization with projection-dimension addresses an *open question* in TRAK [29]: it was unclear why, counter-intuitively, the LDS for TRAK decreases when projection-dimension increases—our result suggests that this is due to an insufficient tuning of regularization. Concretely, Figure 1b and Figure 1c indicate that when having a high projection-dimension (e.g. 8192), TRAK performs poorly with low regularization. However, a higher projection-dimension ultimately leads to better LDS when regularization is tuned.

- **Implicit hyperparameters, such as training-epoch, also play an important role.** Figure 1e illustrates the effect of a largely overlooked hyperparameter, the training-epoch. It turns out that the training-epoch of the model used to calculate the attribution scores could significantly affect LDS. However, it is counter-intuitive that a larger training-epoch could lead to worse LDS, which raises an open question for future research.

## 4 A Case Study on the Regularization Term in Influence Function

In this section, we first analyze the problem of selecting the regularization term $\lambda$ in influence functions for maximizing LDS. Based on our analysis, we propose a surrogate indicator that instructs a practical selection algorithm for $\lambda$ without model retraining. Finally, we evaluate the proposed selection algorithm on a variety of experimental settings, demonstrating that it empirically generalizes well for multiple data attribution methods involving the regularization term.

### 4.1 Problem: Hyperparameter Selection of $\lambda$ in Influence Function

We study the problem of selecting $\lambda$ using the IFFIM version of influence function, $\tau_{\text{IFFIM},\lambda}$, as defined in Eq.(2).

**Hyperparameter selection as an optimization problem.** Recall Definition 2.1, where we have denoted LDS as $c_s(\tau, z', \overline{A})$ for an attributor $\tau$, a test sample $z'$, and samples of subsets $\overline{A}$. The hyperparameter selection problem can be formulated as the following optimization problem:

$$\max_{\lambda > 0} c_s(\tau_{\text{IFFIM},\lambda}, z', \overline{A}).$$

However, directly analyzing this optimization problem poses several technical challenges. First, the LDS objective is non-differentiable because it involves the discrete Spearman correlation. Second, the LDS objective depends on the specific sample of subsets $\overline{A}$, which is difficult to analyze. Third, evaluating the LDS requires model retraining, which we aim to avoid.

**Relaxing the LDS objective.** To address the first two challenges, we propose the Population Pearson LDS as a more tractable proxy for LDS.

**Definition 4.1** (Population Pearson LDS). *For an attributor $\tau$ and a test example $z' \in \mathcal{Z}$, the* Population Pearson LDS *is defined using the Pearson correlation over all subsets $A \subset S$ of size $a$:*

$$c_p(\tau, z') := \text{PearsonCorr}\left( \{f(z', \theta_A^*)\}_{A \subset S, |A|=a}, \Big\{ \sum_{z \in A} \tau(z', z) \Big\}_{A \subset S, |A|=a} \right). \tag{6}$$

Note that Spearman correlation is equivalent to the Pearson correlation of the rank-transformed variables. Therefore, both statistics measure monotonic relationships and are generally expected to be highly correlated [10]. The new objective, Population Pearson LDS, is differentiable and replaces the sampled subsets $\overline{A}$ with all size $a$ subsets of $S$, which makes it easier to analyze. Specifically, we can define the partial derivative of $c_p$ with respect to $\lambda$ as

$$\dot{c}_p(\lambda; z') := \frac{\partial c_p(\tau_{\text{IFFIM},\lambda}, z')}{\partial \lambda}. \tag{7}$$

**Analyzing the partial derivative $\dot{c}_p(\lambda; z')$.** In our hyperparameter sensitivity experiments in Section 3, we observe that, empirically, the LDS with respect to the regularization term $\lambda$ often has a uni-modal and concave shape. This observation turns our attention from the original maximization problem to the analysis of the partial derivative $\dot{c}_p(\lambda; z')$. We first show in the following Lemma 4.2 that our empirical observation about the shape of the LDS as a function of $\lambda$ is not a coincidence.

**Lemma 4.2** (Monotonicity in Small $\lambda$; Informal). *Assuming the test sample $z'$ follows a similar distribution as the training data, under some technical assumptions deferred to Section B.4,[6] with high probability over the choice of test sample $z'$, there exists some $C > 0$ such that for $0 < \lambda < C$,*

$$\dot{c}_p(\lambda; z') > 0.$$

Lemma 4.2 suggests that for a range of small $\lambda$ starting from 0, $c_p$ as a function of $\lambda$ is increasing. With the observation regarding the shape of the LDS curve, ideally, we can solve for the optimal hyperparameter $\lambda^*$ as the stationary point with $\dot{c}_p(\lambda^*; z') = 0$. However, the partial derivative $\dot{c}_p(\lambda^*; z')$ still involves quantities $\theta_A^*$ for $A \subset S$, which require model retraining.

## 4.2 Analysis: A Sufficient Condition for Monotonicity in $\lambda$

Instead of directly solving for the stationary point, we derive a sufficient condition (Theorem 4.3) for the derivative to be positive, which leads to a surrogate indicator that does not require model retraining (see Definition 4.6 in Section 4.3). Before we state this condition, we first introduce some key notations and their intuitive interpretations.

**LOO weighted loss gradients.** We let $D_a$ denote the uniform distribution of size $a$ subsets of $S$. Define $g_{z'} := \frac{1}{n} \sum_{i=1}^{n} \alpha_{z',i} \nabla_\theta L(z_i, \theta_S^*) \in \mathbb{R}^p$, where $\alpha_{z'} \in \mathbb{R}^n$ with $\alpha_{z',i} := \mathbb{E}_{A \sim D_a}[f(z', \theta_A^*)|z_i \in A] - \mathbb{E}_{A \sim D_a}[f(z', \theta_A^*)]$. As the name suggests, we point out that $\alpha_{z',i}$ can be interpreted as the LOO influence of $z_i$ on $z'$ under certain conditions. Under this interpretation, $g_{z'}$ becomes the sum of loss gradients of training points weighted by their LOO influences on the test example $z'$. More details can be found in Section B.2.

**Other necessary notation.** For simplicity of notation in our subsequent analysis, we let $v_{z'}$ denote the gradient $\nabla_\theta f(z', \theta_S^*)$, and define several bilinear terms of $v_{z'}, g_{z'}$, or $\alpha_{z'}$. They will be used to measure and compare different $\theta$ update directions (see Theorem 4.3 and its interpretations below). Denote $t_{k,z',\lambda} := v_{z'}^\top (F_S + \lambda I_p)^{-k} F_S v_{z'}$. Further, we let $r_{z',\lambda} := -v_{z'}^\top (F_S + \lambda I_p)^{-1} g_{z'}$ and $o_{z',\lambda} := \alpha_{z'}^\top (JJ^\top + n\lambda I_n)^{-1} \alpha_{z'}$, where $J \in \mathbb{R}^{n \times p}$ has its $i$th row being $\nabla_\theta L(z_i, \theta_S^*)^\top$. The relationship between $(F_S + \lambda I_p)^{-1}$ and $(JJ^\top + n\lambda I_n)^{-1}$ is embodied in Lemma B.3. Intuitively, these terms define semi-inner products of $v_{z'}, (F_S + \lambda I_p)^{-1} v_{z'}, g_{z'}$ and $\alpha_{z'}$ weighted by matrices related to the FIM.

Now we are ready to state a sufficient condition for the monotonicity of Population Pearson LDS.

**Theorem 4.3** (Sufficient Condition for Positive Derivative). *It suffices for $\dot{c}_p(\lambda; z') > 0$ if*

$$\frac{r_{z',\lambda}}{\sqrt{o_{z',\lambda} \cdot t_{1,z',\lambda}}} > \frac{t_{2,z',\lambda}}{\sqrt{t_{3,z',\lambda} \cdot t_{1,z',\lambda}}}. \tag{8}$$

**Intuition behind Theorem 4.3.** Let's first focus on RHS of Eq. (8). By observing the definition of $t_{k,z',\lambda}$ for $k = 1, 2, 3$, RHS can be viewed as the cosine "angle" between $v_{z'}$ and $(F_S + \lambda I_p)^{-1} v_{z'}$. Here, $(F_S + \lambda I_p)^{-1} v_{z'}$ is related to the $\theta$ update direction in IFFIM, as in Eq. (2). When RHS is small, this "angle" is overly large, suggesting a significant misalignment between the $\theta$ update direction and the gradient descent direction direction. In this case, increasing $\lambda$ helps intuitively because it conditions the inverted matrix $F_S$ better, stabilizing the update direction. When the "angle" is too small, $F_S$ is overshadowed by the regularization term $\lambda I_p$ (as $v_{z'}$ is roughly parallel to $(F_S + \lambda I_p)^{-1} v_{z'}$), meaning that the attribution method may become dominated by regularization and lose sensitivity to the underlying structure of data. Similarly, LHS of Eq. (8) measures the "angle" between $v_{z'}$ and the ground-truth direction that involves $g_{z'}$. Hence Theorem 4.3 is in essence stating that when the "angle" between $v_{z'}$ and $(F_S + \lambda I_p)^{-1} v_{z'}$ is of comparable level as the "angle" between $v_{z'}$ and the ground-truth direction, Population Pearson LDS could reach its maximum.

---

[6]This involves non-trivial technical assumptions that do not always strictly hold. But we have made intuitive and empirical justifications about them.

**Remark 4.4** (Proof Sketch for Theorem 4.3)**.** *The detailed proof can be found in Section B.3. We provide a proof sketch here. We start by expanding $\dot{c}_p(\lambda; z')$ with the definition of $\tau_{\text{IFFIM},\lambda}$. Then, We draw connections between $\text{Var}_{A \sim D_a}[\nabla_\theta R_A(\theta_S^*)]$, a crucial term occurred in the expression of $\dot{c}_p(\lambda; z')$, and $F_S$. Further, we state a condition (Eq. (10)) equivalent to $\dot{c}_p(\lambda; z') > 0$. Finally, we derive Eq. (8) from Eq. (10) with Lemma B.3 and the generalized Cauchy-Schwarz inequality.*

**Remark 4.5** (Gradient Projection)**.** *A result similar to Theorem 4.3 can also be derived for IFFIM attributor with gradient projection (see $\tau_{\text{IFFIM},\lambda,P}$ in Eq. (4)). In this case, given the projection matrix $P$, $t_{k,z',\lambda}$ should be replaced with $v_{z'}^\top P(P^\top F_S P + \lambda I_{\tilde{p}})^{-k} P^\top F_S P P^\top v_{z'}$ while $r_{z',\lambda}$ should be replaced with $-v_{z'}^\top P(P^\top F_S P + \lambda I_{\tilde{p}})^{-1} P^\top g_{z'}$. Additionally, $JJ^\top + n\lambda I_n$ in the $o_{z',\lambda}$ should be replaced with $JPP^\top J^\top + n\lambda I_n$. Please see more details in Section B.5.*

### 4.3 Algorithm: The Surrogate-Indicator-Based Practical Selection

**The surrogate indicator.**  The sufficient condition in Theorem 4.3 enables a practical selection algorithm for $\lambda$ based on the following surrogate indicator $\xi_{z',\lambda}$.

**Definition 4.6** (Surrogate Indicator)**.** *We define the* surrogate indicator *as*

$$\xi_{z',\lambda} := \frac{t_{2,z',\lambda}}{\sqrt{t_{3,z',\lambda} \cdot t_{1,z',\lambda}}}, \tag{9}$$

*which is the RHS of Eq. (8).*

Note that the quantities $t_{k,z',\lambda}$ for $k = 1, 2, 3$ only depends on model parameters $\theta_S^*$ trained on the full dataset $S$, thus $\xi_{z',\lambda}$ can be calculated without retraining. Moreover, both the LHS and the RHS ($\xi_{z',\lambda}$) of Eq. (8) are well normalized.

**Proposition 4.7.** *If $r_{z',\lambda} > 0$, then LHS and RHS of Eq. (8) both lie in the interval $[0,1]$.*

**The practical selection algorithm.**  In principle, we need both the LHS and RHS of Eq. (8) to evaluate the sufficient condition in Theorem 4.3. In practice, however, we find that using $\bar{\xi}_{T,\lambda} := \frac{1}{|T|} \sum_{z' \in T} \xi_{z',\lambda} = 0.5$, where $T$ is a validation dataset, as the criterion[7] to select the regularization term $\lambda$ is uniformly effective across all the experimental settings. This yields a practical algorithm (Algorithm 1) for selecting the hyperparameter $\lambda$ without requiring model retraining.

---

**Algorithm 1** Selecting $\lambda$ with the surrogate indicator.

---

    **Input:** A candidate set $\mathcal{C}$ of $\lambda$, a subset $T \subset \mathcal{Z}$ of test examples.
    **Output:** A selected $\hat{\lambda}$.
1: **for** $\lambda \in \mathcal{C}$ **do**
2:     Compute $\xi_{z',\lambda}$ for all $z' \in T$;
3:     $\bar{\xi}_{T,\lambda} \leftarrow \frac{1}{|T|} \sum_{z' \in T} \xi_{z',\lambda}$;
4: **end for**
5: $\hat{\lambda} \leftarrow \arg\min_{\lambda \in \mathcal{C}} |\bar{\xi}_{T,\lambda} - 0.5|$;

---

### 4.4 Experiments: Validating the Proposed Surrogate-Indicator-Based Selection Algorithm

We present experiments validating the proposed algorithm for selecting the regularization term $\lambda$.

**Experiment settings.**  Dataset, model, and projection dimension: We employ 6 experiment settings, covering different datasets, model scales, and projection dimensions. First two settings apply Logistic Regression models (LR) on MNIST dataset [25], where one utilizes random projection with dimension 512 and the other uses no projection. They are followed by two settings that apply MLP, with random projection 512 and 4096 respectively. Finally, we have two relatively large models, including ResNet-9 [14] on CIFAR-2 dataset [24], and MusicTransformer (MT) [2] on MAESTRO dataset [13]. We fix the random projection dimension to be 4096 in the last two settings. We also conduct experiments

---

[7]In fact, setting the threshold as 0.4 or 0.6 is similarly effective. The $\xi_{z',\lambda}$ as a function of $\log \lambda$ looks like a logistic function and it varies sharply when its value is around 0.5. See Section C.1 for visualizations.

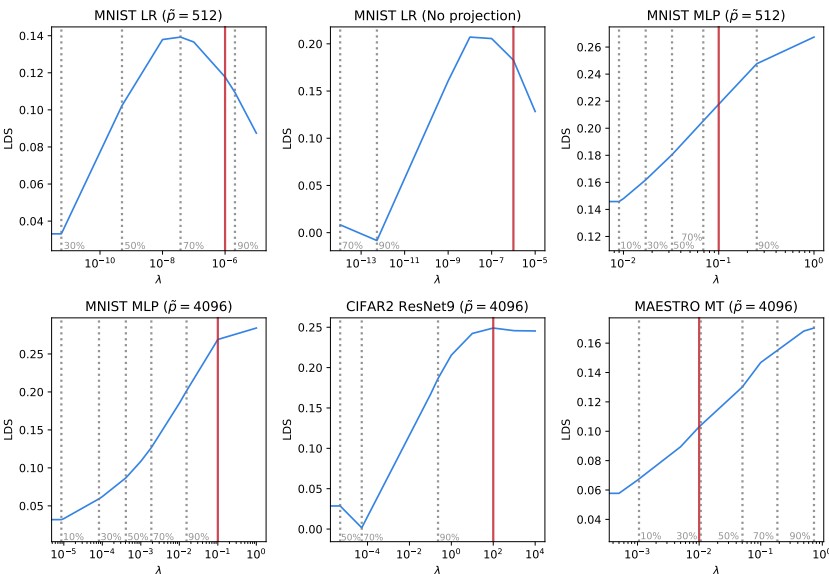

Figure 2: The plot of LDS versus $\lambda$ on the IFFIM attributor. Each subplot corresponds to an experiment setting. The red solid vertical line indicates the $\lambda$ selected by our method. The gray dotted lines indicate the eigenvalues of $F_S$ corresponding to the 10%, 30%, 50%, 70%, and 90% quantiles. The quantiles are annotated in gray color near the x-axis.

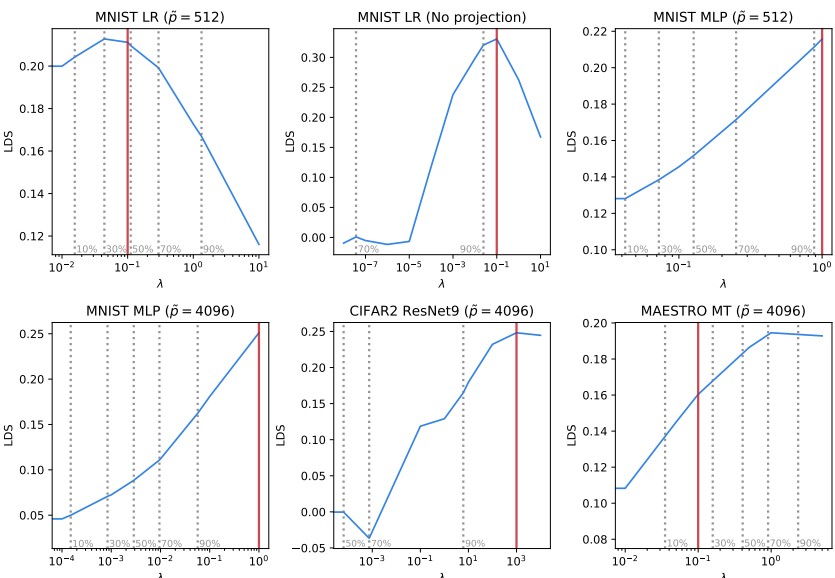

Figure 3: The plot of LDS versus $\lambda$ on the TRAK attributor. Please see Figure 2 for the plotting setup.

on WikiText2 [28] with GPT2 [31], the results of which can be found in Section C.4. Attributor: We experiment with the IFFIM attributor defined in Eq. (2) (or Eq. (4) with projection), and the TRAK attributor [29]. Baseline selection method: We also compare the proposed algorithm against a baseline method that selects $\lambda$ based on the spectrum of $F_S$ since the main motivation of $\lambda$ is to deal with the singularity of $F_S$. Specifically, the baseline method set $\lambda$ to be the eigenvalue of $F_S$ at a fixed quantile among $\{10\%, 30\%, 50\%, 70\%, 90\%\}$.

**Results.** We demonstrate the experimental results on the IFFIM and TRAK respectively in Figure 2 and Figure 3. In each subplot, the blue curve represents the change in LDS with respect to $\lambda$. We mark

the $\lambda$ chosen by the proposed algorithm by the red solid vertical line, and the baselines in the gray dotted vertical lines. Overall, we observe that the $\lambda$ chosen by the proposed algorithm generally leads to good LDS value that is significantly better than $\lambda = 0$, and in many case it is close to the optimal value. Moreover, while the proposed algorithm is based on an analysis of the IFFIM attributor, it generalizes well to the TRAK attributor under various experiment settings with different projection dimensions. In contrast, there is no single fixed quantile for spectrum-based baseline method that consistently performs well, indicating that this naive heuristic is insufficient for selecting $\lambda$.

**Downstream tasks performance.** We also include an evaluation of our proposed algorithm on downstream settings, and the detailed results are presented in Section C.5. We focus on the task of data selection commonly used in data attribution literature [11], where we measure the performance drop of models when the most positively influential training data are removed. We note here that a larger drop signifies stronger attribution quality. In general, we observe that our proposed algorithm achieves visible model performance drops, demonstrating its effectiveness in downstream tasks.

## 5 Discussion and Conclusion

This work brings attention to a fundamental yet overlooked challenge in data attribution: the uniquely high computational cost associated with hyperparameter tuning. Unlike typical machine learning models—where validation metrics can be cheaply computed—evaluating data attribution quality often requires repeated retraining on data subsets, making standard hyperparameter tuning procedures impractical. By systematically characterizing this issue, we contribute the first large-scale empirical study on hyperparameter sensitivity in data attribution methods, and establish this hyperparameter tuning bottleneck as a critical concern for their practical deployment.

Through extensive experiments across diverse methods, datasets, and models, we demonstrate that many data attribution methods are indeed highly sensitive to hyperparameter configurations, with optimal choices varying significantly across settings. For example, we show that some hyperparameters—such as regularization and projection dimension—interact in nontrivial ways, amplifying sensitivity and necessitating more careful tuning strategies. Moreover, implicit factors (such as training epoch) matter; comparing attributions across checkpoints can be informative.

To mitigate the high computational cost of standard evaluation metrics, we focus on the regularization parameter in influence function-based methods and propose a practical selection procedure based on a theoretically motivated surrogate indicator. Our method avoids model retraining and shows robust empirical performance across multiple benchmarks and attributors, making it a practically useful tool for tuning one of the most critical hyperparameters in popular data attribution methods.

**Limitations.** Our theoretical analysis and surrogate-based tuning procedure focus specifically on the regularization parameter in influence function methods. Extending similar ideas to other hyperparameters or to alternative data attribution methods remains an important direction for future research. Moreover, our empirical evaluation primarily relies on the LDS metric. While we believe LDS provides a strong starting point for highlighting the practical challenges of hyperparameter tuning, it would be valuable to investigate whether similar sensitivities arise under alternative evaluation metrics.

## Acknowledgment

The authors would like to thank Juhan Bae for helpful discussions that partially motivated this project.

This project was in part supported by NCSA Delta GPU at NCSA through allocation CIS240402 from the Advanced Cyberinfrastructure Coordination Ecosystem: Services & Support (ACCESS) program [5], which is supported by National Science Foundation grants #2138259, #2138286, #2138307, #2137603, and #2138296. HZ is partially supported by an NSF IIS grant No. 2416897 and a Google Research Scholar Award. The views and conclusions expressed in this paper are solely those of the authors and do not necessarily reflect the official policies or positions of the supporting companies and government agencies.

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

# A  Details of the Hyperparameter Study

## A.1  Hyperparameters Definitions

Apart from major hyperparameters described in Section 2.1, some additional hyperparameters are considered in our hyperparameter sensitivity study. We provide their definitions as following:

- normalization (TracIn) [30]: We may normalize the per-example gradients in TracIn by dividing their L2-norm. In other words, normalization=True means that $\tau_{\text{TracIn}}(z', z_i) := \sum_t \eta_t \frac{\nabla_\theta f(z', \theta_t)^\top}{\|\nabla_\theta f(z', \theta_t)^\top\|} \frac{\nabla_\theta L(z_i, \theta_t)}{\|\nabla_\theta L(z_i, \theta_t)\|}$.

- checkpoint-selection (TracIn) [30]: The checkpoints used to calculate $\tau_{\text{TracIn}}$ can be designed in various way. In this paper, we select the last 10 epochs for different maximum training epoch numbers.

- max-iteration (IF (CG)) [22]: This hyperparameter specifies the maximum number of steps the conjugate gradient algorithm will take to attempt convergence. In the experiment, we do not set stop criteria so that the algorithm will stop at the maximum number of iteration.

- scaling & recursion-depth & batch-size (IF (LiSSA)) [22]: The LiSSA algorithm for inverse Hessian vector product (IHVP) is an iterative algorithm where each iteration $t$ applies the formula $v^t = g + (I - \frac{1}{\eta}(H^t + \lambda I))v^{t-1}$, and $g$ is the target vector. The hyperparameters scaling refers to $\eta$; recursion-depth indicates the maximum $t$ where the iteration stops; batch-size indicates how many data points are used to calculate the batch-wise hessian matrix $H^t$.

**Default values.**  In the following table, we list the default value of each hyperparameter. The default value is used when other hyperparameters are searched. They are selected according to the default value in original paper.

Table 1: Default values of hyperparameters.

| TDA method | Hyperparameter Name | Default values |
|---|---|---|
| TRAK-10 | regularization
pojection-dimension
training-epoch | 0
512 (2048 for WikiText2+GPT2)
50 (3 for WikiText2+GPT2) |
| LoGra | regularization
pojection-dimension
training-epoch | 1e-3
$64^2$
3 |
| IF (explicit) | regularization
training-epoch | 1e-5
50 |
| IF (CG) | regularization
max-iteration
training-epoch | 1e-2
10
50 |
| IF (LiSSA) | regularization
scaling
recursion-depth
batch-size
training-epoch | 1e-3
5
1000
50
50 |
| TracIn | normalization
pojection-dimension
checkpoint-selection | False
None (i.e., no projection)
last 10 checkpoints |

## A.2 Additional Results

In Figure 4, we present additional results not included in Figure 1 due to space limit in the main text. These additional results confirm that most TDA methods are sensitive to certain hyperparameters. Intriguingly, training-epoch is one of the most sensitive hyperparameter for most TDA methods.

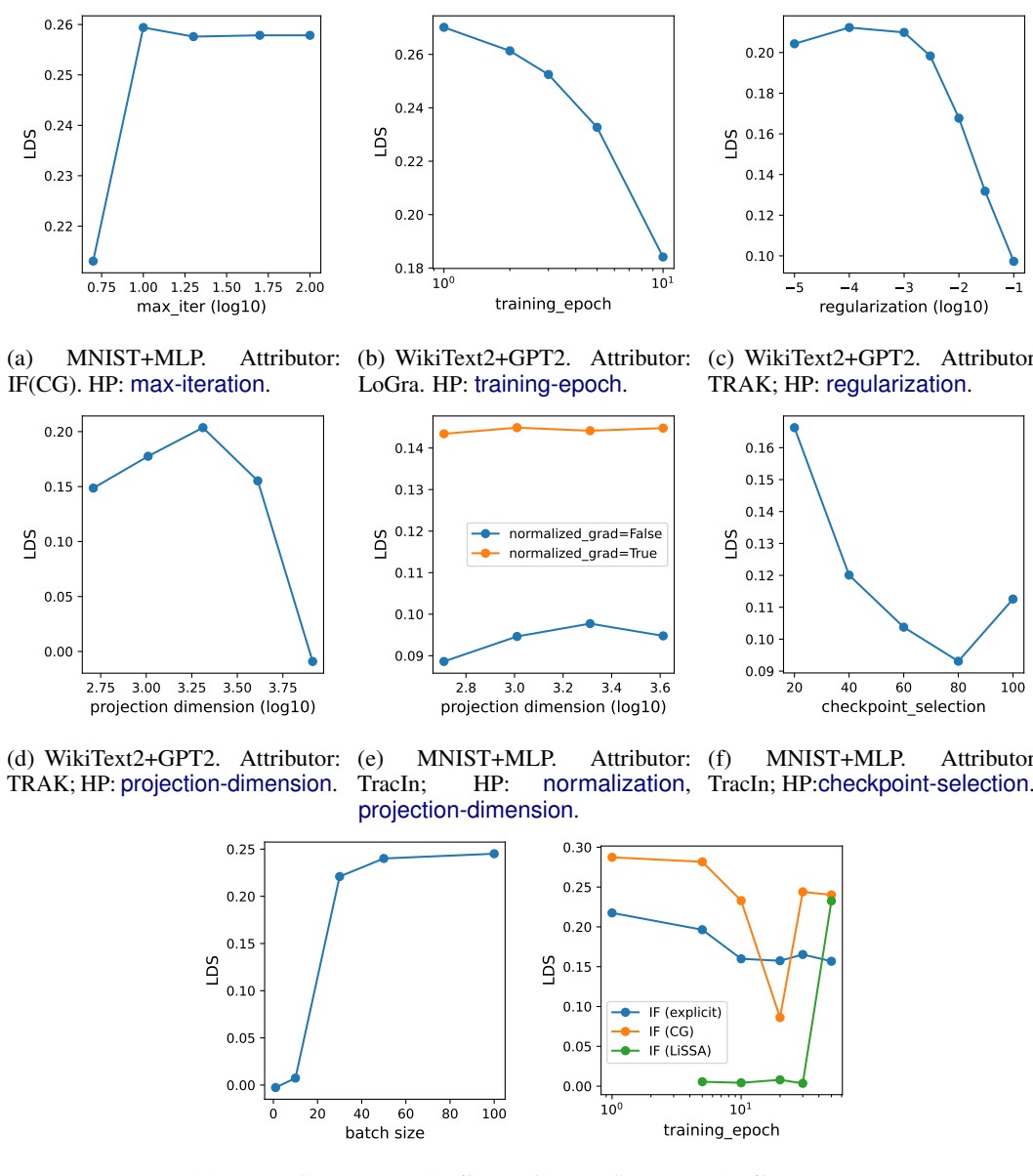

(a) MNIST+MLP. Attributor: IF(CG). HP: max-iteration.

(b) WikiText2+GPT2. Attributor: LoGra. HP: training-epoch.

(c) WikiText2+GPT2. Attributor: TRAK; HP: regularization.

(d) WikiText2+GPT2. Attributor: TRAK; HP: projection-dimension.

(e) MNIST+MLP. Attributor: TracIn; HP: normalization, projection-dimension.

(f) MNIST+MLP. Attributor: TracIn; HP:checkpoint-selection.

(g) MNIST+MLP. Attributor: IF(LiSSA). HP: batch-size.

(h) MNIST+MLP. Attributor: IF. HP: training-epoch.

Figure 4: Study of hyperparameter sensitivity additional results.

## A.3 Computational Resources and Dataset Licenses

The experiments for the hyperparameter sensitivity analysis are done on 4 A100 GPUs in around 100 hours, excluding model retraining (we reused some model checkpoints provided by the dattri library to avoid extensive model retraining). For the dataset we use: MNIST-10 dataset holds CC BY-SA 3.0 license; CIFAR-10 dataset holds CC-BY 4.0 license; WikiText2 dataset holds CC BY-SA 3.0 license.

# B  Omitted Details of the Theoretical Analysis in Section 4

## B.1  Relationship between TRAK and IFFIM

As noted in Section 2.1, TRAK can be viewed as a variant of IFFIM with additional computational tricks. This section demonstrates their similarity and distinctions in more details.

We first formally introduce the TRAK attributor without gradient projection as follows[8].

$$\tau_{\text{TRAK}}(z', z_i) := \nabla_\theta f(z', \theta_S^*)^\top (\Phi^\top R \Phi)^{-1} \nabla_\theta f(z_i, \theta_S^*) \cdot (1 - p_i),$$

where $\Phi \in \mathbb{R}^{n \times p}$ has its $i$th row being $\nabla_\theta f(z_i, \theta_S^*)^\top$, $R := \text{diag}\{p_i(1 - p_i)\}_{i=1}^n$, and $p_i := p(z_i, \theta_S^*)$.

Compared to the IFFIM attributor in Eq. (2), there are two main differences. First, the right most gradient term in TRAK is $\nabla_\theta f(z_i, \theta_S^*)(1 - p_i)$ while its counterpart in IFFIM is $-\nabla_\theta L(z_i, \theta_S^*)$. Second, IFFIM has the empiricla FIM $F_S$ in the middle while TRAK has $\Phi^\top R \Phi$.

**The right gradient term difference.**  We first show that $\nabla_\theta f(z_i, \theta_S^*)(1 - p_i)$ is equivalent to $-\nabla_\theta L(z_i, \theta_S^*)$ as

$$-\nabla_\theta L(z_i, \theta_S^*) = - \left. \frac{\partial \ln(1 + e^{-f})}{\partial f} \right|_{f = f(z_i, \theta_S^*)} \nabla_\theta f(z_i, \theta_S^*) = (1 - p(z_i, \theta_S^*)) \nabla_\theta f(z_i, \theta_S^*).$$

**The middle term difference.**  The middle term in TRAK can be written as

$$\frac{1}{n} \Phi^\top R \Phi = \frac{1}{n} \sum_{i=1}^n \nabla_\theta f(z_i, \theta_S^*)(p_i(1 - p_i)) \nabla_\theta f(z_i, \theta_S^*)^\top.$$

Noting that $p_i(1 - p_i)$ is the Hessian of cross-entropy loss with respect to model output, and the whole $\frac{1}{n} \Phi^\top R \Phi$ is known as the Gerneralized Gauss Newton (GGN) matrix [26].

Furthermore, Martens [26] has shown that $\frac{1}{n} J^\top J$ (the empirical FIM) and $\frac{1}{n} \Phi^\top R \Phi$ (the GGN) both reduce to the true FIM when $S$ converges to the true underlying distribution of $\mathcal{Z}$.

This comparison indicates that there is fundamental similarity between TRAK and IFFIM despite superficial algorithmic differences.

## B.2  LOO Weighted Loss Gradients

As suggested by Park et al. [29], we assume decreasing $a$ by 1 does not shift the distribution of $f(z', \theta_A^*)$, i.e.,

$$\mathbb{E}_{A \sim D_a}[f(z', \theta_A^*)|z_i \notin A] = \mathbb{E}_{A \sim D_{a-1}}[f(z', \theta_A^*)|z_i \notin A].$$

Then, by rewriting the RHS, we have

$$\mathbb{E}_{A \sim D_a}[f(z', \theta_A^*)|z_i \notin A] = \mathbb{E}_{A \sim D_a}[f(z', \theta_{A \setminus \{z_i\}}^*)|z_i \in A],$$

and hence

$$\alpha_{z',i} = \mathbb{E}_{A \sim D_a}[f(z', \theta_A^*)|z_i \in A] - \mathbb{E}_{A \sim D_a}[f(z', \theta_A^*)]$$

$$= \mathbb{E}_{A \sim D_a}[f(z', \theta_A^*)|z_i \in A]$$
$$\quad - \Pr_{A \sim D_a}[z_i \in A]\mathbb{E}_{A \sim D_a}[f(z', \theta_A^*)|z_i \in A] - \Pr_{A \sim D_a}[z_i \notin A]\mathbb{E}_{A \sim D_a}[f(z', \theta_A^*)|z_i \notin A]$$

$$= \Pr_{A \sim D_a}[z_i \notin A](\mathbb{E}_{A \sim D_a}[f(z', \theta_A^*)|z_i \in A] - \mathbb{E}_{A \sim D_a}[f(z', \theta_A^*)|z_i \notin A])$$

$$= (1 - \frac{a}{n})\mathbb{E}_{A \sim D_a}[f(z', \theta_A^*) - f(z', \theta_{A \setminus \{z_i\}}^*)|z_i \in A],$$

which means that the weights $\alpha_{z',i}$ in $g_{z'}$ can be interpreted as the LOO influence of $z_i$ on $z'$.

---

[8]In practical implementation, Park et al. [29] dropped the diagonal matrix $R$ due to slightly improved empirical performance, and projected $\Phi$ and the gradients to lower dimension for computational efficiency.

## B.3 Proof of Theorem 4.3

We prove Theorem 4.3 before Lemma 4.2 for better readability. We first establish several intermediate results for Eq. (8).

**Lemma B.1.** *We have*

$$\mathrm{Var}_{A\sim D_a}[\nabla_\theta R_A(\theta_S^*)] = \frac{n-a}{a(n-1)}F_S,$$

*where $D_a$ is defined in Section 4.2.*

*Proof.* By the optimality of $\theta_S^*$, we know $\sum_{z\in S}[\nabla_\theta L(z,\theta_S^*)] = n\cdot\nabla_\theta R_S(\theta_S^*) = 0$. We also have $\mathbb{E}_{A\sim D_a}[\sum_{z\in A}\nabla_\theta L(z,\theta_S^*)] = 0$.

Now, because $A$ is uniformly sampled size $a$ subsets of $S$,

$$\mathrm{Var}_{A\sim D_a}[\nabla_\theta R_A(\theta_S^*)] = \mathrm{Var}_{A\sim D_a}[\frac{1}{a}\sum_{z\in A}\nabla_\theta L(z,\theta_S^*)]$$

$$= \frac{1}{a^2}\mathbb{E}_{A\sim D_a}[(\sum_{z\in A}\nabla_\theta L(z,\theta_S^*))(\sum_{z\in A}\nabla_\theta L(z,\theta_S^*))^\top] - 0$$

$$= \frac{1}{a^2}(a\mathbb{E}_{z\sim S}[\nabla_\theta L(z,\theta_S^*)\nabla_\theta L(z,\theta_S^*)^\top] + \mathbb{E}_{A\sim D_a}[\sum_{\substack{z_1\in A\\z_2\in A\\z_1\neq z_2}}\nabla_\theta L(z_1,\theta_S^*)\nabla_\theta L(z_2,\theta_S^*)^\top]).$$

The second summand can be reduced to

$$\mathbb{E}_{A\sim D_a}[\sum_{\substack{z_1\in A\\z_2\in A\\z_1\neq z_2}}\nabla_\theta L(z_1,\theta_S^*)\nabla_\theta L(z_2,\theta_S^*)^\top])$$

$$= \frac{1}{\binom{n}{a}}\sum_{A\in D_a}\sum_{\substack{z_1\in A\\z_2\in A\\z_1\neq z_2}}\nabla_\theta L(z_1,\theta_S^*)\nabla_\theta L(z_2,\theta_S^*)^\top$$

$$= \frac{1}{\binom{n}{a}}\sum_{\substack{z_1\in S\\z_2\in S\\z_1\neq z_2}}\binom{n-2}{a-2}\nabla_\theta L(z_1,\theta_S^*)\nabla_\theta L(z_2,\theta_S^*)^\top$$

$$= \frac{a(a-1)}{n(n-1)}\sum_{\substack{z_1\in S\\z_2\in S\\z_1\neq z_2}}\nabla_\theta L(z_1,\theta_S^*)\nabla_\theta L(z_2,\theta_S^*)^\top$$

$$= \frac{a(a-1)}{n(n-1)}((\sum_{z\in S}\nabla_\theta L(z,\theta_S^*))(\sum_{z\in S}\nabla_\theta L(z,\theta_S^*))^\top - \sum_{z\in S}\nabla_\theta L(z,\theta_S^*)\nabla_\theta L(z,\theta_S^*)^\top)$$

$$= -\frac{a(a-1)}{n-1}\mathbb{E}_{z\sim S}[\nabla_\theta L(z,\theta_S^*)\nabla_\theta L(z,\theta_S^*)^\top],$$

where $A\in D_a$ means $A$ is a size $a$ subset of $S$.

Finally, by the definition of $F_S$,

$$\mathrm{Var}_{A\sim D_a}[\nabla_\theta R_A(\theta_S^*)] = \frac{1}{a^2}(a\cdot F_S - \frac{a(a-1)}{n-1}\cdot F_S) = \frac{n-a}{a(n-1)}F_S.$$

$\square$

Lemma B.1 facilitates analyzing the variance of $\sum_{z\in A}\tau_{\mathrm{IFFIM},\lambda}(z',z)$ in Eq. (6), thereby enabling verification of the following equivalent condition of $\dot{c}_p(\lambda;z') > 0$.

**Proposition B.2.** *For a test example $z'$, $\dot{c}_p(\lambda; z') > 0$ is equivalent to*

$$r_{z',\lambda} \cdot t_{3,z',\lambda} > (-\nabla_\theta f(z', \theta_S^*)^\top (F_S + \lambda I_p)^{-2} g_{z'}) \cdot t_{2,z',\lambda}. \tag{10}$$

*Proof.* We first simplify $\sum_{z \in A} \tau_{\text{IFFIM},\lambda}(z', z)$ in Eq. (6):

$$\sum_{z \in A} \tau_{\text{IFFIM},\lambda}(z', z) = -\sum_{z \in A} \nabla_\theta f(z', \theta_S^*)^\top (F_S + \lambda I_p)^{-1} \nabla_\theta L(z, \theta_S^*)$$

$$= -a \cdot \nabla_\theta f(z', \theta_S^*)^\top (F_S + \lambda I_p)^{-1} \nabla_\theta R_A(\theta_S^*).$$

Three terms are involved when we expand the definition of $c_p(\tau_{\text{IFFIM},\lambda})$. The first is the numerator:

$$\mathbb{E}_{A \sim D_a}[\sum_{z \in A} \tau_{\text{IFFIM},\lambda}(z', z) \cdot (f(z', \theta_A^*) - \mathbb{E}_{A' \sim D_a}[f(z', \theta_{A'}^*)])]$$

$$= \mathbb{E}_{A \sim D_a}[-a\nabla_\theta f(z', \theta_S^*)^\top (F_S + \lambda I_p)^{-1} \nabla_\theta R_A(\theta_S^*) \cdot (f(z', \theta_A^*) - \mathbb{E}_{A' \sim D_a}[f(z', \theta_{A'}^*)])],$$

where the component that depends on $A$ is

$$\mathbb{E}_{A \sim D_a}[\nabla_\theta R_A(\theta_S^*) \cdot (f(z', \theta_A^*) - \mathbb{E}_{A' \sim D_a}[f(z', \theta_{A'}^*)])]$$

$$= \frac{1}{\binom{n}{a}} \sum_{A \in D_a} (f(z', \theta_A^*) - \mathbb{E}_{A' \sim D_a}[f(z', \theta_{A'}^*)]) \frac{1}{a} \sum_{z \in A} \nabla_\theta L(z, \theta_S^*)$$

$$= \frac{1}{a\binom{n}{a}} \sum_{i=1}^n (\sum_{A \in D_a} [\![z_i \in A]\!](f(z', \theta_A^*) - \mathbb{E}_{A' \sim D_a}[f(z', \theta_{A'}^*)])) \nabla_\theta L(z_i, \theta_S^*)$$

$$= \frac{1}{a\binom{n}{a}} \sum_{i=1}^n \binom{n-1}{a-1} \mathbb{E}_{A \sim D_a}[f(z', \theta_A^*) - \mathbb{E}_{A' \sim D_a}[f(z', \theta_{A'}^*)]|z_i \in A] \nabla_\theta L(z_i, \theta_S^*)$$

$$= \frac{1}{n} \sum_{i=1}^n \mathbb{E}_{A \sim D_a}[f(z', \theta_A^*) - \mathbb{E}_{A' \sim D_a}[f(z', \theta_{A'}^*)]|z_i \in A] \nabla_\theta L(z_i, \theta_S^*)$$

$$= \frac{1}{n} \sum_{i=1}^n \alpha_{z',i} \nabla_\theta L(z_i, \theta_S^*) = g_{z'}.$$

Note that we apply $a\binom{n}{a} = n\binom{n-1}{a-1}$.

The second term is the variance of $\sum_{z \in A} \tau_{\text{IFFIM},\lambda}(z', z)$:

$$\text{Var}_{A \sim D_a}[\sum_{z \in A} \tau_{\text{IFFIM},\lambda}(z', z)]$$

$$= \text{Var}_{A \sim D_a}[-a \cdot \nabla_\theta f(z', \theta_S^*)^\top (F_S + \lambda I_p)^{-1} \nabla_\theta R_A(\theta_S^*)]$$

$$= a^2 \cdot \nabla_\theta f(z', \theta_S^*)^\top (F_S + \lambda I_p)^{-1} \text{Var}_{A \sim D_a}[\nabla_\theta R_A(\theta_S^*)](F_S + \lambda I_p)^{-1} \nabla_\theta f(z', \theta_S^*)$$

$$= \frac{a(n-a)}{n-1} \cdot \nabla_\theta f(z', \theta_S^*)^\top (F_S + \lambda I_p)^{-1} F_S (F_S + \lambda I_p)^{-1} \nabla_\theta f(z', \theta_S^*),$$

where we apply the identity Lemma B.1.

The third term is the variance of $f(z', \theta_A^*)$, which, together with $a$ in the numerator and $\frac{a(n-a)}{n-1}$ in the variance of $\sum_{z \in A} \tau_{\text{IFFIM},\lambda}(z', z)$, are omitted because they do not depend on $\lambda$. To summarize, so far we have shown that $\dot{c}_p(\lambda; z') > 0$ is equivalent to

$$\frac{\partial}{\partial \lambda} \frac{-\nabla_\theta f(z', \theta_S^*)^\top (F_S + \lambda I_p)^{-1} g_{z'}}{\sqrt{\nabla_\theta f(z', \theta_S^*)^\top (F_S + \lambda I_p)^{-1} F_S (F_S + \lambda I_p)^{-1} \nabla_\theta f(z', \theta_S^*)}} \Bigg|_\lambda > 0.$$

By direct calculation, the left hand side is

$$\frac{(-\nabla_\theta f(z', \theta_S^*)^\top (F_S + \lambda I_p)^{-1} g_{z'})(\nabla_\theta f(z', \theta_S^*)^\top (F_S + \lambda I_p)^{-3} F_S \nabla_\theta f(z', \theta_S^*))}{- (-\nabla_\theta f(z', \theta_S^*)^\top (F_S + \lambda I_p)^{-2} g_{z'})(\nabla_\theta f(z', \theta_S^*)^\top (F_S + \lambda I_p)^{-2} F_S \nabla_\theta f(z', \theta_S^*))}{(\nabla_\theta f(z', \theta_S^*)^\top (F_S + \lambda I_p)^{-1} F_S (F_S + \lambda I_p)^{-1} \nabla_\theta f(z', \theta_S^*))^{3/2}}.$$

When the positive denominator is dropped, we obtain the desired formula.

$\square$

Before we move to the proof of next result, we first introduce a useful matrix transformation lemma.

**Lemma B.3.** *For matrix $X \in \mathbb{R}^{n \times p}$, non-negative integer $k$, and $\lambda > 0$, we have*

$$(X^\top X + \lambda I_p)^{-k} X^\top = X^\top (X X^\top + \lambda I_n)^{-k}.$$

*Proof.* Note that

$$X^\top (X X^\top + \lambda I_n) = X^\top X X^\top + \lambda X^\top = (X^\top X + \lambda I_p) X^\top.$$

If we left multiply $(X^\top X + \lambda I_p)^{-1}$ and right multiply $(X X^\top + \lambda I_n)^{-1}$ on both sides, we derive

$$(X^\top X + \lambda I_p)^{-1} X^\top = X^\top (X X^\top + \lambda I_n)^{-1}.$$

By applying the equality $k$ times on $(X^\top X + \lambda I_p)^{-k} X^\top$ to "push $X^\top$ through" the inverted matrix, we complete the proof. $\square$

*Proof of Theorem 4.3.* Because $F_S$ is a positive semi-definite matrix, so is $(F_S + \lambda I_p)^{-k} F_S$ for any positive integer $k$. As a result,

$$t_{k,z',\lambda} = \nabla_\theta f(z', \theta_S^*)^\top (F_S + \lambda I_p)^{-k} F_S \nabla_\theta f(z', \theta_S^*) \geq 0.$$

In fact, $t_{k,z',\lambda} = 0 \Leftrightarrow F_S \nabla_\theta f(z', \theta_S^*) = 0$ implies $t_{2,z',\lambda} = 0$ which results in an undefined $c_p$. Therefore, $t_{k,z',\lambda} > 0$. Further, by the premise, we have

$$r_{z',\lambda} > \frac{t_{2,z',\lambda}}{\sqrt{t_{3,z',\lambda} \cdot t_{1,z',\lambda}}} \sqrt{o_{z',\lambda} \cdot t_{1,z',\lambda}} > 0.$$

Without loss of generality, we only consider the case where $-\nabla_\theta f(z', \theta_S^*)^\top (F_S + \lambda I_p)^{-2} g_{z'} > 0$, because otherwise Eq. (10) holds automatically as its left hand side would be positive while right hand side would be non-positive, which guarantees $\dot{c}_p(\lambda; z') > 0$ by Proposition B.2.

We proceed by rewriting Eq. (10) with Lemma B.3. By setting $X = J/\sqrt{n}$, we have

$$t_{3,z',\lambda} = \frac{1}{n} \nabla_\theta f(z', \theta_S^*)^\top J^\top (\frac{1}{n} J J^\top + \lambda I_n)^{-3} J \nabla_\theta f(z', \theta_S^*),$$

$$-\nabla_\theta f(z', \theta_S^*)^\top (F_S + \lambda I_p)^{-2} g_{z'} = -\frac{1}{n} \nabla_\theta f(z', \theta_S^*)^\top J^\top (\frac{1}{n} J J^\top + \lambda I_n)^{-2} \alpha_{z'},$$

where we apply the fact that $F_S = \frac{1}{n} J^\top J = X^\top X$ and $g_{z'} = \frac{1}{n} J^\top \alpha_{z'}$ by their definitions. Then, note that

$$
\begin{aligned}
&\sqrt{t_{3,z',\lambda} \cdot o_{z',\lambda}} \\
&= \sqrt{(\frac{1}{n} \nabla_\theta f(z', \theta_S^*)^\top J^\top (\frac{1}{n} J J^\top + \lambda I_n)^{-3} J \nabla_\theta f(z', \theta_S^*))(\frac{1}{n} \alpha_{z'}^\top (\frac{1}{n} J J^\top + \lambda I_n)^{-1} \alpha_{z'})} \quad (11) \\
&\geq -\frac{1}{n} \nabla_\theta f(z', \theta_S^*)^\top J^\top (\frac{1}{n} J J^\top + \lambda I_n)^{-2} \alpha_{z'} = -\nabla_\theta f(z', \theta_S^*)^\top (F_S + \lambda I_p)^{-2} g_{z'} > 0,
\end{aligned}
$$

by applying the generalized Cauchy-Schwarz inequality on $-(\frac{1}{n} J J^\top + \lambda I_n)^{-1} J \nabla_\theta f(z', \theta_S^*)$ and $\alpha_{z'}$ with inner product defined by positive definite matrix $\frac{1}{n}(\frac{1}{n} J J^\top + \lambda I_n)^{-1}$. Now, we finish the proof by observing that Eq. (10) is derived directly by multiplying Eq. (8) with Eq. (11) and rearranging terms. $\square$

## B.4 Assumptions and Proof of Lemma 4.2

We base the discussion of Lemma 4.2 on Eq. (8). For simplicity, we first eliminate $\sqrt{t_{1,z',\lambda}}$ on both sides to state the following sufficient condition for $\dot{c}_p(\lambda; z') > 0$:

$$\frac{r_{z',\lambda}}{\sqrt{o_{z',\lambda}}} > \frac{t_{2,z',\lambda}}{\sqrt{t_{3,z',\lambda}}}. \tag{12}$$

Now, for $i = 1, 2, \ldots, \max\{n, p\}$, we let $\mu_i$ denote the $i$th largest eigenvalue of $F_S$ if $i \leq p$, and $0$ otherwise. We have the following result for $\mu_i$:

**Lemma B.4.** *For $i = 1, 2, \ldots, n$, $\mu_i$ is the $i$th largest eigenvalue of $\frac{1}{n} J J^\top$.*

*Proof.* Write the singular value decomposition $J = U \Sigma V^\top$ with $\Sigma_{ii} = \sigma_i$ for $i \in [\min\{n, p\}]$. As $F_S = \frac{1}{n} J^\top J = \frac{1}{n} V \Sigma^\top \Sigma V^\top$, we know $\sigma_i = \sqrt{n \cdot \mu_i}$ for $i \in [\min\{n, p\}]$. Further, $\frac{1}{n} \Sigma \Sigma^\top = \text{diag}\{[\![i \leq \min\{n, p\}]\!] \frac{1}{n} \sigma_i^2\}_{i=1}^n = \text{diag}\{\mu_i\}_{i=1}^n$ because $\mu_i = 0$ for $i > p$. We finish the proof by noting that $\frac{1}{n} J J^\top = \frac{1}{n} U \Sigma \Sigma^\top U^\top$. $\square$

To better establish a relationship between the spectrum of $F_S$ and $c_p(\tau_{\text{IFFIM}, \lambda})$, we first make the following definition for convenience.

**Definition B.5.** *For a vector $v \in \mathbb{R}^p$ and $i \in [\max\{n, p\}]$, let $\tilde{v}_i$ denote the $i$th component of $v$ under a chosen eigenbasis of $F_S$, where eigenvalues are sorted in descending order if $i \leq p$; $\tilde{v}_i = 0$ otheriwise. Further, let $\tilde{v}_{\min}$ denote the component of $v$ under the eigenbasis corresponding to the minimal non-zero eigenvalue $\mu_{\min}$ of $F_S$.*

We then provide a result on the expectation of $\widetilde{\nabla_\theta L}_i(z, \theta_S^*)^2$, the square of the $i$th eigen-component of $\nabla_\theta L(z, \theta_S^*)$, when $z$ ranges over $S$.

**Lemma B.6.** *For $i = 1, 2, \ldots, p$, we have*

$$\mathbb{E}_{z \sim S}[\widetilde{\nabla_\theta L}_i(z, \theta_S^*)^2] = \mu_i.$$

*Proof.* We adapt the proof from [6]. Let's write the eigen-decomposition of $F_S$:

$$F_S = V \Lambda V^\top$$

where $\Lambda = \text{diag}\{\mu_i\}_{i=1}^p$. Then

$$\begin{aligned}
\Lambda &= V^\top F_S V \\
&= V^\top \mathbb{E}_{z \sim S}[\nabla_\theta L(z, \theta_S^*) \nabla_\theta L(z, \theta_S^*)^\top] V \\
&= \mathbb{E}_{z \sim S}[V^\top \nabla_\theta L(z, \theta_S^*) \nabla_\theta L(z, \theta_S^*)^\top V] \\
&= \mathbb{E}_{z \sim S}\left[\left[\widetilde{\nabla_\theta L}_i(z, \theta_S^*) \widetilde{\nabla_\theta L}_j(z, \theta_S^*)\right]_{i,j=1}^p\right].
\end{aligned}$$

We obtain the desired equality by comparing diagonal terms. $\square$

Next, we introduce an assumption regarding the concentration of distribution of $\widetilde{\nabla_\theta L}_i(z', \theta_S^*)^2$ when the test example $z'$ is sampled.

**Assumption B.7.** *Assume there exist constants $0 < C_1 < C_2$ such that for $i = 1, 2, \ldots, n$,*

$$C_1 \mathbb{E}_{z \sim S}[\widetilde{\nabla_\theta L}_i(z, \theta_S^*)^2] \leq \widetilde{\nabla_\theta L}_i(z', \theta_S^*)^2 \leq C_2 \mathbb{E}_{z \sim S}[\widetilde{\nabla_\theta L}_i(z, \theta_S^*)^2], \tag{13}$$

*with high probability over the sample of $z'$ from the test distribution.*

Assumption B.7 basically assumes that the distribution given by $S$ represents the test distribution well regarding the relative size of eigen-components. Then, we are able to derive an upper bound for the RHS (right hand side) of Eq. (12).

**Proposition B.8.** *Under Assumption B.7, for all $\lambda > 0$, we have*

$$\text{RHS} < \frac{nC_2}{(1 - p(z', \theta_S^*))\sqrt{C_1}} \frac{(\mu_{\min} + \lambda)^{3/2}}{\mu_{\min}}, \tag{14}$$

*with high probability over the sample of $z'$ from the test distribution.*

*Proof.* It is known that the $i$th component of $J\nabla_\theta f(z', \theta_S^*)$ under the eigenbasis of $\frac{1}{n}JJ^\top$ is precisely

$$\sqrt{n \cdot \mu_i} \widetilde{\nabla_\theta f}_i(z', \theta_S^*) = \frac{\sqrt{n \cdot \mu_i}}{p(z', \theta_S^*) - 1} \widetilde{\nabla_\theta L}_i(z', \theta_S^*),$$

for $i = 1, 2, \ldots, n$. This can be shown by employing the singular value decomposition $J = U\Sigma V^\top$. Under the eigenbasis of $\frac{1}{n}JJ^\top$,

$$U^\top J\nabla_\theta f(z', \theta_S^*) = \Sigma V^\top \nabla_\theta f(z', \theta_S^*),$$

so the identity holds because $\Sigma_{ii} = \sqrt{n \cdot \mu_i}$ for $i \in [\min\{n, p\}]$ (see Lemma B.4) and $\mu_i = 0$ for $i > p$.

For the numerator, with high probability,

$$
\begin{aligned}
t_{2,z',\lambda} &= \frac{1}{n}\nabla_\theta f(z', \theta_S^*)^\top J^\top (\frac{1}{n}JJ^\top + \lambda I_n)^{-2} J\nabla_\theta f(z', \theta_S^*) \\
&= \frac{1}{n(1 - p(z', \theta_S^*))^2} \sum_{i=1}^n \frac{(\sqrt{n \cdot \mu_i}\widetilde{\nabla_\theta L}_i(z', \theta_S^*))^2}{(\mu_i + \lambda)^2} \\
&\leq \frac{1}{(1 - p(z', \theta_S^*))^2} \sum_{i=1}^n \frac{C_2\mu_i^2}{(\mu_i + \lambda)^2} \\
&< \frac{1}{(1 - p(z', \theta_S^*))^2} \sum_{i=1}^n [\![\mu_i \neq 0]\!]C_2 \\
&\leq \frac{n}{(1 - p(z', \theta_S^*))^2}C_2.
\end{aligned}
$$

For the denominator, with high probability,

$$
\begin{aligned}
t_{3,z',\lambda} &= \frac{1}{n}\nabla_\theta f(z', \theta_S^*)^\top J^\top (\frac{1}{n}JJ^\top + \lambda I_n)^{-3} J\nabla_\theta f(z', \theta_S^*) \\
&= \frac{1}{n(1 - p(z', \theta_S^*))^2} \sum_{i=1}^n \frac{(\sqrt{n \cdot \mu_i}\widetilde{\nabla_\theta L}_i(z', \theta_S^*))^2}{(\mu_i + \lambda)^3} \\
&\geq \frac{1}{(1 - p(z', \theta_S^*))^2} \sum_{i=1}^n \frac{C_1\mu_i^2}{(\mu_i + \lambda)^3} \\
&\geq \frac{1}{(1 - p(z', \theta_S^*))^2} \frac{C_1\mu_{\min}^2}{(\mu_{\min} + \lambda)^3}.
\end{aligned}
$$

We finish the proof by combining these two bounds. $\qquad\square$

We move to analyze the LHS (left hand side) of Eq.(12). Before that, we need two additional technical assumptions.

**Assumption B.9.** *Assume there exist constants $C_3 > 0$ and $\varepsilon_g > 0$ such that for $i = 1, 2, \ldots, n$,*

$$\tilde{g}_{z',i}^2 \leq C_3\mu_i^{1+\varepsilon_g}. \tag{15}$$

**Remark B.10.** *Because $g_{z'} = \frac{1}{n}J^\top\alpha_{z'}$, for $\mu_i$ equal to $0$, $\tilde{g}_{z',i}$ must also be $0$, which can be easily shown through the singular value decomposition of $J$. Hence, Assumption B.9 holds when $C_3$ is large enough. However, we note here that the sufficient condition for $\dot{c}_p(\lambda; z') > 0$ becomes tighter when $C_3$ is larger and $\varepsilon_g$ is smaller (see Lemma B.15 for details).*

**Assumption B.11.** *Assume $\alpha_{z'}$ is in the column space of $J$.*

**Remark B.12.** *A special case for this assumption is when $J$ has rank $n-1$, where the deducted $1$ rank is because $\sum_{i=1}^{n} \nabla_\theta L(z_i, \theta_S^*) = 0$. In this case, since $\sum_{i=1}^{n} \alpha_{z',i} = 0$ by simple computation, $\alpha_{z'}$ is in the column space of $J$.*

**Lemma B.13.** *Assume Assumption B.11 holds. For $i \in [n]$, if $\mu_i = 0$, then the $i$th component of $\alpha_{z'}$ under the eigenbasis of $\frac{1}{n} JJ^\top$ is also zero.*

*Proof.* Write the singular value decomposition $J = U\Sigma V^\top$. Let $\alpha_{z'} = J\beta$ for some $\beta \in \mathbb{R}^p$. Since $\frac{1}{n} JJ^\top = \frac{1}{n} U\Sigma\Sigma^\top U^\top$, the $i$th component of $\alpha_{z'}$ under the eigenbasis of $\frac{1}{n} JJ^\top$ is

$$(U^\top \alpha_{z'})_i = (U^\top U\Sigma V^\top \beta)_i = (\Sigma V^\top \beta)_i,$$

which is zero for $\mu_i = 0$ by Lemma B.4. $\qquad\square$

Now we are ready to derive an upper bound for $o_{z',\lambda}$.

**Proposition B.14.** *Under Assumption B.7, Assumption B.9, and Assumption B.11, for all $\lambda > 0$,*

$$o_{z',\lambda} \le \frac{C_3 n}{\mu_{\min}^{1-\varepsilon_g}}. \tag{16}$$

*Proof.* Denote $w_{z',i}$ as the $i$th component of $\alpha_{z'}$ under the eigenbasis of $\frac{1}{n} JJ^\top$ for $i \in [n]$. We first show that for $i \in [n]$, if $\mu_i = 0$ then $w_{z',i} = 0$; if $\mu_i \neq 0$, (then $i \le p$ by definition of $\mu_i$) $w_{z',i} = \tilde{g}_{z',i}\sqrt{n/\mu_i}$. For $\mu_i = 0$, Lemma B.13 guarantees that the component of $\alpha_{z'}$ is 0. For $\mu_i \neq 0$, with the singular value decomposition of $J$ in Lemma B.4,

$$\tilde{g}_{z',i} = (V^\top \frac{1}{n} V^\top \Sigma^\top U^\top \alpha_{z'})_i = \frac{1}{n}\sigma_i w_{z',i} = w_{z',i}\sqrt{\frac{\mu_i}{n}},$$

for $i \in [\min\{n,p\}]$, which gives the desired result. Therefore,

$$o_{z',\lambda} = \frac{1}{n}\alpha_{z'}^\top (\frac{1}{n} J^\top J + \lambda I_n)^{-1}\alpha_{z'} = \sum_{i=1}^{n} [\![\mu_i \neq 0]\!]\frac{\tilde{g}_{z',i}^2/\mu_i}{\mu_i + \lambda} \le \sum_{i=1}^{n}[\![\mu_i \neq 0]\!]\frac{C_3 \mu_i^{\varepsilon_g}}{\mu_i + \lambda} \le \frac{C_3 n}{\mu_{\min}^{1-\varepsilon_g}}.$$

$\square$

Finally, by combining all the results, we state the following formal version of Lemma 4.2.

**Lemma B.15** (Formal version of Lemma 4.2). *Assume Assumption B.7, Assumption B.9, and Assumption B.11 hold. With high probability over the sample of $z'$ from the test distribution, $r_{z',0^+} := \lim_{\lambda \to 0^+} r_{z',\lambda}$ exists. Further assume $r_{z',0^+} > 0$. Then if $\mu_{\min} < C_\mu$ where $C_\mu$ is some positive value depending on $z'$, there exists some $C > 0$ such that for $0 < \lambda < C$,*

$$\dot{c}_p(\lambda; z') > 0.$$

*Proof.* We first show that with high probability the limit exists. By expanding $r_{z',\lambda} = -\nabla_\theta f(z', \theta_S^*)^\top (F_S + \lambda I_p)^{-1} g_{z'}$ under the eigenbasis of $F_S$, with high probability, the absolute value of the summation term corresponding to $\mu_i$ for $i \in [\min\{n,p\}]$ is

$$\left|\frac{-\widetilde{\nabla_\theta f}_i(z',\theta_S^*)\tilde{g}_{z',i}}{\mu_i + \lambda}\right| \le \frac{\sqrt{C_2 C_3}}{1 - p(z', \theta_S^*)}\frac{\mu_i^{1/2 + (1+\varepsilon_g)/2}}{\mu_i + \lambda} \le \frac{\sqrt{C_2 C_3}}{1 - p(z', \theta_S^*)}\mu_i^{\varepsilon_g/2}.$$

Additionally, for $i > n$, $\tilde{g}_{z',i} = 0$ from the singular value decomposition of $J$. As a result, $\lim_{\lambda \to 0^+} r_{z',\lambda}$ exists. As $r_{z',0^+} > 0$, there exists $C^* > 0$ such that for all $0 < \lambda < C^*$, $r_{z',\lambda} > \frac{1}{2}r_{z',0^+}$. Now, let

$$C_\mu := \left(\frac{r_{z',0^+}(1 - p(z', \theta_S^*))\sqrt{C_1}}{2nC_2\sqrt{n \cdot C_3}}\right)^{2/\varepsilon_g} > 0,$$

and

$$C := \min\{C^*, (\frac{r_{z',0^+}(1 - p(z', \theta_S^*))\sqrt{C_1}}{2nC_2\sqrt{n \cdot C_3}})^{2/3}\mu_{\min}^{1-\varepsilon_g/3} - \mu_{\min}\}.$$

By direct calculation, if $\mu_{\min} < C_\mu$, then $C > 0$. Further, when $0 < \lambda < C$,

$$\text{LHS} \geq \frac{r_{z',\lambda}}{\sqrt{n \cdot C_3}}\mu_{\min}^{(1-\varepsilon_g)/2} > \frac{r_{z',0^+}}{2\sqrt{n \cdot C_3}}\mu_{\min}^{(1-\varepsilon_g)/2} > \frac{nC_2}{(1 - p(z', \theta_S^*))\sqrt{C_1}}\frac{(\mu_{\min} + \lambda)^{3/2}}{\mu_{\min}} > \text{RHS},$$

which implies $\dot{c}_p(\lambda; z') > 0$. $\qquad\square$

**Remark B.16.** *Our analysis relies on the condition that $\mu_{\min}$, the smallest non-zero eigenvalue of $F_S$, remains small. This assumption aligns with established analyses demonstrating eigenvalue concentration near zero for both the FIM and Hessian in deep neural networks near convergence* [32, 20]. *These results empirically justify our treatment of $\mu_{\min}$.*

We subsequently focus on a discussion of the positivity condition $r_{z',0^+} > 0$ which constitutes the remainder of this section.

**Discussion of $r_{z',0^+} > 0$.** Here we show that, in the special case where $z' = z_i \in S$ for some $i \in [n]$, we have $r_{z',0^+} > 0$. We write the singular value decomposition $J = U\Sigma V^\top$. Then by Lemma B.4,

$$\begin{aligned}
\lim_{\lambda \to 0^+} r_{z',\lambda} &= \lim_{\lambda \to 0^+} -\frac{1}{n}\nabla_\theta f(z', \theta_S^*)^\top J^\top(\frac{1}{n}JJ^\top + \lambda I_n)^{-1}\alpha_{z'} \\
&= \lim_{\lambda \to 0^+} \frac{1}{n(1 - p_i)}\nabla_\theta L(z_i, \theta_S^*)^\top J^\top(\frac{1}{n}JJ^\top + \lambda I_n)^{-1}\alpha_{z'} \\
&= \frac{1}{1 - p_i}\lim_{\lambda \to 0^+} e_i^\top \frac{1}{n}JJ^\top(\frac{1}{n}JJ^\top + \lambda I_n)^{-1}\alpha_{z'} \\
&= \frac{1}{1 - p_i}\lim_{\lambda \to 0^+} \sum_{j=1}^n \frac{\mu_j}{\mu_j + \lambda}(U^\top e_i)_j(U^\top \alpha_{z'})_j \\
&= \frac{1}{1 - p_i}\sum_{j=1}^n [\![\mu_j \neq 0]\!](U^\top e_i)_j(U^\top \alpha_{z'})_j,
\end{aligned}$$

where $e_i$ is the $i$th standard basis vector. By Lemma B.13, $\mu_j = 0$ implies $(U^\top \alpha_{z'})_j = 0$. Therefore,

$$r_{z',0^+} = \lim_{\lambda \to 0^+} r_{z',\lambda} = \frac{1}{1 - p_i}e_i^\top UU^\top \alpha_{z'} = \frac{\alpha_{z',i}}{1 - p_i} > 0,$$

because $p_i < 1$ and $\alpha_{z',i} = \alpha_{z_i,i} = \mathbb{E}_{A \sim D_a}[f(z_i, \theta_A^*)|z_i \in A] - \mathbb{E}_{A \sim D_a}[f(z_i, \theta_A^*)]$ is the expected change in model output of $z_i$ itself when $z_i$ is included in the training set, which is positive.

**Empirical assessment of constants in assumptions.** The results derived above are dependent on the constants $C_1$, $C_2$, and $C_3$ introduced in Assumptions B.7 and B.9. Here we provide a brief discussion on the empirical behavior of these constants, showing that they generally stay around the level of 0.1 to 10 instead of scaling with the smallest eigenvalue $\mu_{\min}$. Note that, in this empirical study, we compute $C_1$ with respect to the smallest eigenvalue rather than all eigenvalues, which is reasonable because the only $C_1$ term that occurs in the theoretical derivation is the one related to $\mu_{\min}$. Further, due to computational limitations, we consider projection dimension 4096. We fix $\varepsilon_g = 0.5$.

To empirically estimate these constants, we first diagonalize $P^\top F_S P$ to obtain its eigenvalues and execute a data attribution pass with IFFIM to compute $\nabla_\theta L(z', \theta_S^*)$ and $g_{z'}$ for $z'$ in the test dataset. Then, we follow Eq.(13) and Eq.(15) to obtain values of $C_1$, $C_2$, and $C_3$ that satisfy these inequalities for all or a given percentage of test examples. For ResNet-9 [14] on CIFAR-2 dataset [24], we find that $C_3 = 0.4082$ is enough for Eq.(15) to hold on all the test examples, and $C_1 = 0.3121$ and $C_2 = 51.9933$ satisfy Eq.(13) on 90% of the test examples. In comparison, the smallest eigenvalue is only about $10^{-4}$. For MusicTransformer [2] on MAESTRO dataset [13], we find that $C_3 = 0.0156$ is enough for all test examples, and 90% of test examples yield $C_1 = 0.0518$ and $C_2 = 21.5589$. In comparison, the smallest eigenvalue is only about $7 \times 10^{-5}$. These results suggest that, even in highly non-convex settings, our assumptions tend to hold empirically.

## B.5 Effects of Gradient Projection

Our derivation does not rely on specific properties of $J$. When incorporating gradient projection via Eq. (4), as outlined in Remark 4.5, the projection matrix $P$ can be systematically applied to all gradient terms. We now analyze how gradient projection affects $\mu_{\min}$, the smallest non-zero eigenvalue of $F_S$ (equivalently, of $\frac{1}{n}JJ^\top$). Under projection, this eigenvalue corresponds to the smallest non-zero eigenvalue of $\frac{1}{n}JPP^\top J^\top$.

In practical applications, gradient projection approximately preserves the dominant eigenvalues of $F_S$. Consequently, when $F_S$ exhibits vanishingly small (but non-zero) eigenvalues near index $\min\{n, \tilde{p}\}$, the projected matrix $\frac{1}{n}JPP^\top J^\top$ retains a comparably small non-zero eigenvalue, validating the critical condition in Lemma B.15.

Unlike Arnoldi-based method [33] that aims to preserve top eigenvalues by design, we resort to the standard subspace embedding results [36] for random projection based methods [6, 29]: For common choices of $P$ (e.g. Gaussian random projection) and $0 < \varepsilon_{\mathrm{rp}}, \delta < 1$, if $\tilde{p} = \Theta((n + \ln(1/\delta))\varepsilon_{\mathrm{rp}}^{-2})$, then with probability $1 - \delta$, $P$ satisfies that for any $v \in \mathbb{R}^n$,

$$(1 - \varepsilon_{\mathrm{rp}})\|J^\top v\|_2 \leq \|P^\top J^\top v\|_2 \leq (1 + \varepsilon_{\mathrm{rp}})\|J^\top v\|_2.$$

This implies, by min-max theorem,

$$(1 - \varepsilon_{\mathrm{rp}})\mu_i(JJ^\top) \leq \mu_i(JPP^\top J^\top) \leq (1 + \varepsilon_{\mathrm{rp}})\mu_i(JJ^\top),$$

where $\mu_i(\cdot)$ stands for the $i$th biggest eigenvalue. This guarantees an approximation of top eigenvalues.

## B.6 Proof of Proposition 4.7

*Proof of Proposition 4.7.* We utilize Lemma B.3 to obtain

$$t_{k,z',\lambda} = \frac{1}{n}\nabla_\theta f(z', \theta_S^*)^\top J^\top (\frac{1}{n}JJ^\top + \lambda I_n)^{-k} J\nabla_\theta f(z', \theta_S^*),$$

$$r_{z',\lambda} = -\frac{1}{n}\nabla_\theta f(z', \theta_S^*)^\top J^\top (\frac{1}{n}JJ^\top + \lambda I_n)^{-1}\alpha_{z'},$$

where $k = 1, 2, 3$. Further,

$$o_{z',\lambda} = \frac{1}{n}\alpha_{z'}^\top (\frac{1}{n}JJ^\top + \lambda I_n)^{-1}\alpha_{z'}.$$

We point out that LHS of Eq. (8) lying in $[0, 1]$ is the direct result of applying the generalized Cauchy-Schwarz inequality on $-J\nabla_\theta f(z', \theta_S^*)$ and $\alpha_{z'}$ with inner product defined by positive definite matrix $\frac{1}{n}(\frac{1}{n}JJ^\top + \lambda I_n)^{-1}$, assuming $r_{z',\lambda} > 0$. Similarly, the RHS of Eq. (8) is shown bounded when the inequality is applied on $-(\frac{1}{n}JJ^\top + \lambda I_n)^{-1}J\nabla_\theta f(z', \theta_S^*)$ and $-J\nabla_\theta f(z', \theta_S^*)$ with the same inner product. We use the fact that $t_{2,z',\lambda} \geq 0$ as $(\frac{1}{n}JJ^\top + \lambda I_n)^{-2}$ is positive semi-definite. $\qquad\square$

# C Details and Extra Experiments of the Surrogate Indicator

## C.1 Visualization of the Average Surrogate Indicator

To provide better insights about the proposed surrogate indicator, we present the curve of $\bar{\xi}_{T,\lambda}$ as a function of $\lambda$ in Figure 5. In all experiment settings, empirically, $\bar{\xi}_{T,\lambda}$ is a monotonic function of $\lambda$ with a *transitional phase* near $\bar{\xi}_{T,\lambda} = 0.5$, indicating a good sensitivity to $\lambda$ around this value, and supporting our choice of threshold in Algorithm 1.

## C.2 Computational Resources and Dataset Licenses

The experiments for the surrogate indicator are done on an A40 GPU in around 10 hours, excluding model retraining (we reused some model checkpoints provided by the dattri library to avoid extensive model retraining). For the datasets we use: MNIST-10 dataset holds CC BY-SA 3.0 license; CIFAR-10 dataset holds CC-BY 4.0 license; MAESTRO dataset holds CC BY-NC-SA 4.0 license.

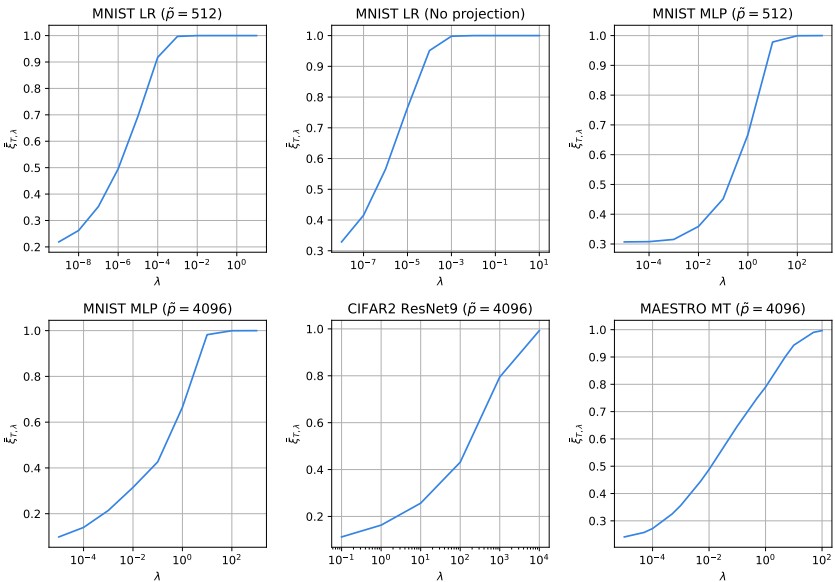

Figure 5: The curve of $\bar{\xi}_{T,\lambda}$ as a function of $\lambda$. Each subfigure corresponds to an experiment setting outlined in Section 4.4.

## C.3 Effects of Different Subset Fractions

We provide an extended analysis of how the subset fraction $a/|S|$ influences the surrogate indicator, considering values in $\{0.25, 0.5, 0.75\}$, where $0.5$ corresponds to the setting used in earlier experiments. Figure 6 illustrates the surrogate indicator's behavior for each subset fraction, shown in green $(0.25)$, blue $(0.5)$, and violet $(0.75)$. While the magnitude of the LDS varies with the subset fraction, its overall trend as a function of $\lambda$ remains similar across different values of subset fraction. This consistency suggests that the surrogate indicator is robust to changes in subset size across different experimental settings.

## C.4 Additional Experiments on WikiText2 with GPT2

Here, we demonstrate additional experiments on the large setting WikiText2 [28] with GPT2 [31]. We fix projection dimension $4096$, and apply the surrogate indicator to find proper regularization strengths. We investigate both the IFFIM and the LoGra [6] attributors. The results are illustrated in Figure 7.

## C.5 Experiments on Downstream Settings

We investigate various settings, including Logistic Regression models (LR) on MNIST dataset [25], ResNet-9 [14] on CIFAR-2 [24] dataset, and MusicTransformer (MT) [2] on MAESTRO dataset [13]. We conduct the experiments with training data removal rate $10\%$, $30\%$, and $50\%$. For methods we use for data removal, we have **Random** (random removal), **IFFIM Default** and **TRAK Default** (IFFIM and TRAK attributors with default regularization 0), and **IFFIM Selected** and **TRAK Selected** (IFFIM and TRAK attributors with selected regularization by our algorithm). For reference, we also include **Full**, which stands for test performance without removal. We repeat the experiments with 10 different seeds and report the mean values as well as their standard errors for better statistical significance. The results are presented in Table 2, Table 3, and Table 4. We notice that when removing $10\%$ of the training data, most methods lead to insignificant model performance decrease, likely due to redundancy in the datasets. However, we observe that our methods (**IFFIM Selected** and **TRAK Selected**) generally outperform baselines (**Random**, **IFFIM Default**, and **TRAK Default**) across different settings and the differences are statistically significant.

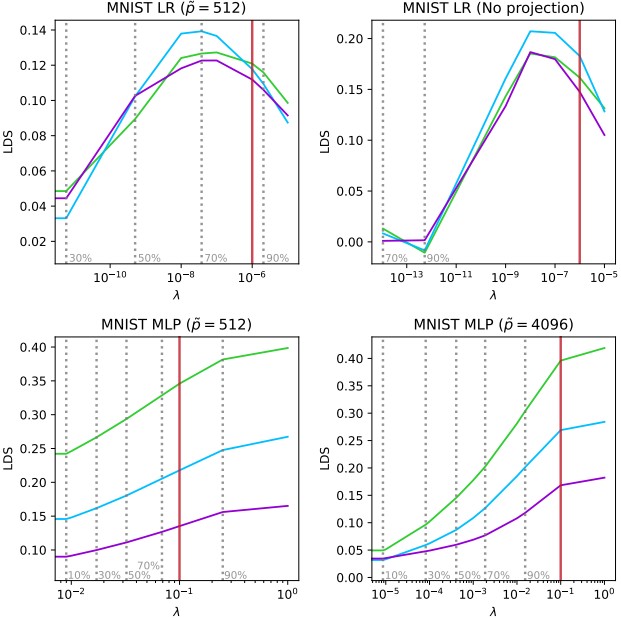

Figure 6: Experiment results with different values of subset fraction. Green, blue, and violet curves illustrate LDS-$\lambda$ relationship with subset fraction $0.25$, $0.5$, and $0.75$, respectively.

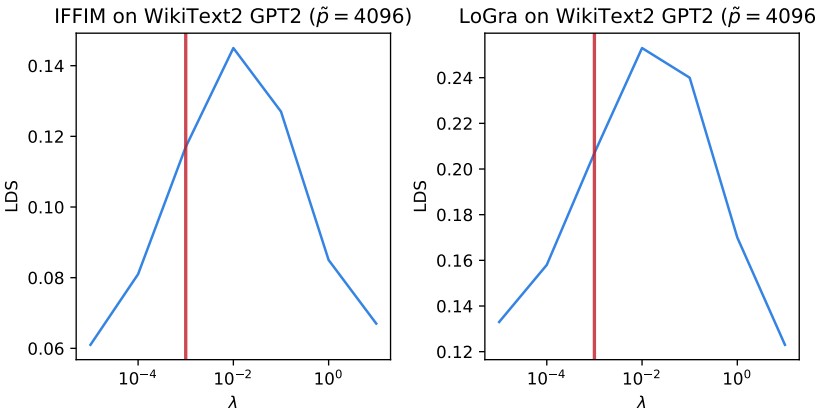

Figure 7: The plot of LDS versus $\lambda$ on the WikiText2 with GPT2 setting. The left plot shows the results with the IFFIM attributor while the right plot demonstrates the results with the LoGra attributor. The red solid vertical line indicates the $\lambda$ selected by our method.

## D   Code Availability

Our code is publicly available at https://github.com/TRAIS-Lab/data-attribution-hp.

Table 2: Data selection performance on MNIST+LR settings. Values in the table are means and standard errors of test accuracies. Experiments are repeated with 10 different seeds. Lower accuracies indicate better attribution performance.

| Removal rates | 10% | 30% | 50% |
|---|---|---|---|
| **Full** | $88.63\% \pm 0.03\%$ | $88.63\% \pm 0.03\%$ | $88.63\% \pm 0.03\%$ |
| **Random** | $88.52\% \pm 0.03\%$ | $88.06\% \pm 0.05\%$ | $87.60\% \pm 0.06\%$ |
| **IFFIM Default** | $88.44\% \pm 0.05\%$ | $87.77\% \pm 0.08\%$ | $87.58\% \pm 0.10\%$ |
| **IFFIM Selected** | $87.66\% \pm 0.04\%$ | $84.50\% \pm 0.04\%$ | $81.53\% \pm 0.06\%$ |
| **TRAK Default** | $88.53\% \pm 0.06\%$ | $87.92\% \pm 0.11\%$ | $86.88\% \pm 0.29\%$ |
| **TRAK Selected** | $\mathbf{87.30\% \pm 0.04\%}$ | $\mathbf{83.84\% \pm 0.05\%}$ | $\mathbf{80.12\% \pm 0.08\%}$ |

Table 3: Data selection performance on CIFAR-2+ResNet-9 settings. Values in the table are means and standard errors of test accuracies. Experiments are repeated with 10 different seeds. Lower accuracies indicate better attribution performance.

| Removal rates | 10% | 30% | 50% |
|---|---|---|---|
| **Full** | $75.04\% \pm 0.53\%$ | $75.04\% \pm 0.53\%$ | $75.04\% \pm 0.53\%$ |
| **Random** | $74.20\% \pm 0.59\%$ | $71.86\% \pm 0.33\%$ | $68.64\% \pm 0.38\%$ |
| **IFFIM Default** | $74.36\% \pm 0.38\%$ | $72.42\% \pm 0.47\%$ | $67.84\% \pm 1.16\%$ |
| **IFFIM Selected** | $74.36\% \pm 0.38\%$ | $72.42\% \pm 0.47\%$ | $67.84\% \pm 1.16\%$ |
| **TRAK Default** | $74.06\% \pm 0.52\%$ | $71.68\% \pm 0.40\%$ | $69.30\% \pm 0.49\%$ |
| **TRAK Selected** | $\mathbf{71.24\% \pm 0.52\%}$ | $\mathbf{64.14\% \pm 0.47\%}$ | $\mathbf{56.30\% \pm 0.70\%}$ |

Table 4: Data selection performance on MAESTRO+MT settings. Values in the table are means and standard errors of test losses. Experiments are repeated with 10 different seeds. Higher losses indicate better attribution performance. Note that we only include TRAK here, as IF has been shown to exhibit poor attribution performance in this complex setting [8].

| Removal rates | 10% | 30% | 50% |
|---|---|---|---|
| **Full** | $4.30 \pm 0.01$ | $4.30 \pm 0.01$ | $4.30 \pm 0.01$ |
| **Random** | $4.32 \pm 0.01$ | $4.38 \pm 0.01$ | $4.67 \pm 0.02$ |
| **TRAK Default** | $4.35 \pm 0.01$ | $4.40 \pm 0.01$ | $4.68 \pm 0.01$ |
| **TRAK Selected** | $\mathbf{4.40 \pm 0.01}$ | $\mathbf{4.52 \pm 0.01}$ | $\mathbf{4.83 \pm 0.02}$ |

