# OpenReview forum: "Taming Hyperparameter Sensitivity in Data Attribution: Practical Selection Without Costly Retraining"
_NeurIPS.cc/2025/Conference — NeurIPS 2025 poster_

### Official Review · Reviewer_GgCL · 2025-06-29

**Clarity:** 4
**Significance:** 2
**Originality:** 2
**Rating:** 5
**Confidence:** 4

**Summary:**

The paper investigates hyperparameters for data attribution and benchmarks their impact on several prominent methods: Influence Functions, TraceIN, TRAK, and LoGra, across the MNIST (MLP), CIFAR-10 (ResNet-9), and Wikitext (GPT-2) datasets. The main findings can be summarized as follows:

- Most attribution methods are sensitive to certain hyperparameters.
- The optimal hyperparameter choices vary across datasets and models.
- The impact of hyperparameters can be entangled.
- Implicit hyperparameters, such as training epochs, also play a significant role.

Finally, the authors propose an algorithm (surrogate-indicator-based practical selection) for selecting the regularization term for influence functions without the need for retraining. The method is validated on MNIST, CIFAR, and Maestro datasets, showing that “overall, we observe that the λ chosen by the proposed algorithm generally leads to good LDS values that are significantly better than λ = 0, and in many cases, it is close to the optimal value.”

**Questions:**

Are there any findings or insights that are more surprising, or could any of the results be suggested as actionable steps for practitioners?

**Ethical Concerns:**

["NO or VERY MINOR ethics concerns only"]

**Final Justification:**

The underexplored role of hyperparameters in TDA: evidenced by the limited support in existing libraries (e.g. OpenDataVal) makes this paper more relevant than I first thought.
If hyperparameters in TDA are underexamined, then insights I initially deemed marginal (e.g. the impact of training epoch) gain in importance.
The new GPT-2 experiments with LDS and, especially, the downstream tasks (in response to Reviewer GpMe) convincingly show that the regularization term substantially improves data-removal performance. With the default parameter, results are near random; with the tuned parameter (especially for TRAK) the improvement is clear. This downstream evaluation more effectively demonstrates the method’s benefits than the earlier benchmarks in my opinion.

**Limitations:**

yes

**Quality:**

3

**Strengths And Weaknesses:**

Strengths: The paper is easy to follow and addresses relevant questions that are analyzed thoroughly.

Weaknesses: I do not find the contributions significant enough for NeurIPS. The insights from the study are not particularly surprising and may not provide substantial benefits to a broad set of researchers; the finding that “the best hyperparameter choices vary across datasets and models” indicates that researchers and practitioners will still need to perform hyperparameter searches after reading the paper. While the algorithm assists in selecting one parameter for influence functions, it focuses on a single parameter for one method, and I am not convinced of its generalizability to other datasets. In my opinion, this paper would be better suited for a specialized venue.

---

> ### Author Rebuttal · Authors · 2025-07-31
>
> We thank the reviewer for the comments and questions. We address your individual concerns below.
>
> > Are there any findings or insights that are more surprising
>
> We would like to highlight that our paper has **several novel/surprising findings with practical implications**. These findings are discussed in detail in Section 2 and 3. For instance, we demonstrate that parameters like *training epoch* **have NOT been considered as hyperparameters in prior literature**, but they significantly affect the performance of a wide range of data attribution methods. As another example, by studying the entanglement of hyperparameters, we have provided a compelling explanation for an **open question raised in the TRAK paper**: the counterintuitive LDS drop with respect to increasing projection dimension can be counteracted by introducing a proper regularization. In short, our work extends **beyond** the observation that hyperparameter selection is highly case-specific, which is true, but also shed light on concrete behaviors of attribution performance when hyperparameters are tuned with evidence from the first systematic research in this direction.
>
> > or could any of the results be suggested as actionable steps for practitioners
>
> We kindly refer the reviewer to the end of Section 1 and Section 5 for some discussion on the implications of our results. Specifically, (1) we highlighted that hyperparameter tuning is a unique challenge in data attribution given the high computational cost associated with the evaluation metrics—to the best of our knowledge, ours is the first study to identify this unique challenge. (2) We pointed out that future methodological development of data attribution should have a more systematic treatment on hyperparameter selection. (3) Our proposed method in Section 4 provides both *a practical method* for selecting the regularization term in many popular data attribution methods (including IF, TRAK, LoGra), and *an example* for analyzing the hyperparameters.
>
> In addition, some of our novel findings in Section 3 have straightforward implications for practitioners. For example, when using random projection in gradient-based methods, one should carefully tune the regularization term (possibly with our proposed method).
>
> In our next revision, we will include a dedicated paragraph to discuss these implications in one place.
>
> > it focuses on a single parameter for one method, and I am not convinced of its generalizability to other datasets
>
> We would like to clarify that our surrogate indicator algorithm applies to a wide range of settings instead of one method or a limited number of datasets. As we discuss in Section 2 and Appendix, state-of-the-art data attribution methods that involve inverting FIM (or GGN) and introduce random projection can be categorized as a variant of our proposed IFFIM, which include but are not limited to TRAK and LoGra. Our discussion of IFFIM can be easily generalized to these methods. Further, following the reviewer’s suggestion, we include an additional larger experimental setting on text data (GPT-2 on WikiText-2), which supports the effectiveness of our algorithm to select a good regularization coefficient. The results are summarized in the table below.
> | Attributor/Regularization | 1e-5  | 1e-4  | 1e-3             | 1e-2  | 1e-1  | 1e0   | 1e1   |
> |---------------------------|-------|-------|------------------|-------|-------|-------|-------|
> | IFFIM                     | 0.061 | 0.081 | 0.117 (selected) | 0.145 | 0.127 | 0.085 | 0.067 |
> | LoGra                     | 0.133 | 0.158 | 0.207 (selected) | 0.253 | 0.240 | 0.170 | 0.123 |
> Here, (selected) indicates the regularization strength selected by our algorithm. The performances are measured in LDS.
>
> We highlight that our study now covers experiments on **three different modalities** (image, text, and music), as well as **three types of data attribution methods** (IFFIM, TRAK, and LoGra). We are actively expanding our experimental coverage and will include additional results in the revised version to further support generalizability.

---

> > ### Comment · Reviewer_GgCL · 2025-08-06
> >
> > Thank you for your response and the additional clarifications. After reconsidering, I will raise my rating to “marginal accept.” My main motivations are:
> >
> > - The underexplored role of hyperparameters in TDA: evidenced by the limited support in existing libraries (e.g. OpenDataVal) makes this paper more relevant than I first thought.
> > - If hyperparameters in TDA are underexamined, then insights I initially deemed marginal (e.g. the impact of training epoch) gain in importance.
> > - The new GPT-2 experiments with LDS and, especially, the downstream tasks (in response to Reviewer GpMe) convincingly show that the regularization term substantially improves data-removal performance. With the default parameter, results are near random; with the tuned parameter (especially for TRAK) the improvement is clear. This downstream evaluation more effectively demonstrates the method’s benefits than the earlier benchmarks in my opinion.
> >
> > I recommend a marginal accept mainly because the experimental validation could be strengthened by:
> > - Reporting how much data is removed at each removal step.
> > - Showing performance across different removal levels (e.g. 10%, 20%, 30%).
> >
> > Finally, since the default regularization parameter is set to 0, could a nonzero default be more suitable?

---

> > > ### Author Response · Authors · 2025-08-08
> > >
> > > Thank you for your time and we appreciate your reconsideration! We respond to the additional questions below.
> > >
> > > > Reporting how much data is removed at each removal step. Showing performance across different removal levels (e.g. 10%, 20%, 30%).
> > >
> > > We are a bit unsure what "each removal step" refers to as in each of our settings, we just did one-time removal.
> > >
> > > Per reviewer's request, we add several different removal levels to demonstrate the capability of our method. During the process, we notice that when removing 10% of the data, none of the methods leads to significant performance decrease, likely because MNIST and CIFAR have certain data redundancy. In light of this observation and to better demonstrate the statistical significance, we repeat the experiment with 10 different seeds and report mean values as well as their standard errors in the table below.
> > >
> > > | Settings           | Full         | R            | ID           | IS           | TD           | TS           |
> > > |--------------------|--------------|--------------|--------------|--------------|--------------|--------------|
> > > | MNIST+LR+10%       | 88.63 ± 0.03 | 88.52 ± 0.03 | 88.44 ± 0.05 | 87.66 ± 0.04 | 88.53 ± 0.06 | 87.30 ± 0.04 |
> > > | MNIST+LR+30%       | 88.63 ± 0.03 | 88.06 ± 0.05 | 87.77 ± 0.08 | 84.50 ± 0.04 | 87.92 ± 0.11 | 83.84 ± 0.05 |
> > > | MNIST+LR+50%       | 88.63 ± 0.03 | 87.60 ± 0.06 | 87.58 ± 0.10 | 81.53 ± 0.06 | 86.88 ± 0.29 | 80.12 ± 0.08 |
> > > | CIFAR2+ResNet9+10% | 75.04 ± 0.53 | 74.20 ± 0.59 | 74.36 ± 0.38 | 74.36 ± 0.38 | 74.06 ± 0.52 | 71.24 ± 0.52 |
> > > | CIFAR2+ResNet9+30% | 75.04 ± 0.53 | 71.86 ± 0.33 | 72.42 ± 0.47 | 72.42 ± 0.47 | 71.68 ± 0.40 | 64.14 ± 0.47 |
> > > | CIFAR2+ResNet9+50% | 75.04 ± 0.53 | 68.64 ± 0.38 | 67.84 ± 1.16 | 67.84 ± 1.16 | 69.30 ± 0.49 | 56.30 ± 0.70 |
> > >
> > > Here the percentage at the end of the settings denotes the removal rates, and our previous results are based on a 50% removal rate. As shown in the table, in both 30% and 50% cases, our methods (IS and TS) generally outperform baselines (R, ID, and TD) and the difference is statistically significant, while there is not much difference across different methods in the 10% setting, likely due to data redundancy as we discussed above.
> > >
> > > > Finally, since the default regularization parameter is set to 0, could a nonzero default be more suitable?
> > >
> > > While it is possible to set a nonzero default value, it is tricky to find a universally satisfactory default value since the best values vary across different settings, as observed and discussed in our empirical study in the paper. For example, comparing (c) and (f) in Figure 1, the best regularization value in (f) for the red top line is $10^{-2}$, while this is the worst regularization value for the purple bottom line in (c). Our hyperparameter search method provides a theoretically justified way to adaptively set the regularization parameter and mitigates this challenge.

---

### Official Review · Reviewer_GpMe · 2025-07-01

**Clarity:** 3
**Significance:** 3
**Originality:** 2
**Rating:** 4
**Confidence:** 4

**Summary:**

This paper focuses on the problem of hyperparameter selection in data attribution methods, which is a topic that is severely underestimated in existing research but is crucial in practical applications. The authors first conduct a systematic large-scale empirical study to evaluate the hyperparameter sensitivity of mainstream data attribution methods on multiple models and datasets. In addition, the authors propose a surrogate indicator based on theoretical derivation, which can select the regularization hyperparameter λ without retraining the model. Experiments show that the indicator has good consistency and performance in multiple datasets and methods.

**Questions:**

Overall, I appreciate the contribution of this paper, so I gave it a positive rating. My main questions are as follows:
1. In addition to λ, the projection dimension and the number of training rounds have also been shown to have a significant impact on performance. Why didn't you try to estimate them using a surrogate-like approach? Is it because of mathematical difficulty, non-differentiability, or other practical obstacles?

2. Have you considered other performance indicators? All current experiments use LDS as the only metric. Have you considered using downstream performance as an evaluation metric in data denoising or data selection tasks? If it cannot be extended to practical tasks, the practicality of this paper will be questioned.

3. Are theoretical indicators generalizable? The derivation of the surrogate is based on a series of technical assumptions. Do these assumptions hold true in large-scale deep models or strong non-convex situations? It is recommended to provide further empirical support or hypothesis sensitivity analysis.

4. Is there a risk of "misleading selection"? The author's method attempts to fit the LDS trend with the surrogate curve, but is there a situation where the surrogate indicator fails to be monotonically in some areas, resulting in the misselection of suboptimal λ? Have the authors considered uncertainty estimates or confidence bounds to aid in the selection?

**Ethical Concerns:**

["NO or VERY MINOR ethics concerns only"]

**Final Justification:**

After reading the rebuttal and discussion, I feel the authors have addressed my main concerns, especially by adding downstream experiments and clarifying theoretical assumptions. Some limitations remain in scope and complexity, but the strengths still outweigh the weaknesses, so I keep my borderline accept recommendation.

**Limitations:**

Yes

**Quality:**

3

**Strengths And Weaknesses:**

Strengths
1. This paper focuses on the practical issues in the implementation of data attribution methods.

2. This paper covers multiple attribution methods and is highly representative.

3. Attached with GitHub links and detailed experimental details, it is helpful for the community to verify and use it later.

Weaknesses
1. The proposed surrogate indicator is only applicable to the λ parameter in the influence function method, and cannot be extended to other key hyperparameters (such as projection dimension) or other types of attribution methods, with obvious limitations.

2. The experimental indicators are single, and many results are based on the synthetic indicator LDS. End-to-end verification has not been performed in actual tasks (such as noise detection and training data screening), and it cannot be proved that the selected hyperparameters can bring better downstream performance.

3. The proposed indicators are based on complex partial derivative conditions and high-order tensor expressions, which lack interpretability for non-theoretical researchers and have a high threshold for actual adoption.

4. The author identified the pain points of hyperparameter tuning, but the solution he finally gave is more like an "ad hoc fix" rather than a systematic solution to the problem.

---

> ### Author Rebuttal · Authors · 2025-07-31
>
> We appreciate the positive evaluation as well as the constructive feedback given by the reviewer. We address each point in Weaknesses and Questions below.
>
> > The proposed surrogate indicator is only applicable to the λ parameter in the influence function method, and cannot be extended to other key hyperparameters (such as projection dimension) or other types of attribution methods, with obvious limitations.
> > In addition to λ, the projection dimension and the number of training rounds have also been shown to have a significant impact on performance. Why didn't you try to estimate them using a surrogate-like approach? Is it because of mathematical difficulty, non-differentiability, or other practical obstacles?
>
> We would like to clarify that the regularization strength is one of the most influential and sensitive hyperparameters according to our empirical study, which justifies our special attention to it. Regularization strength is also applicable to a wide range of state-of-the-art attribution methods, including but not limited to TRAK and LoGra (as discussed in Section 2 and Appendix). Additionally, we would like to point out some practical difficulties when applying similar surrogate indicator analysis on other hyperparameters such as projection dimension and training epoch. While we include some discussion of gradient projection and its influence on the effectiveness of the surrogate indicator in Section 4, projection dimension is difficult to analyze directly, as its impact is highly dependent on the choice of random projection matrices. Similarly, to predict the effect of tuning training epochs, we have to carefully consider a variety of external factors such as optimization methods and learning rate, which complicates theoretical modeling and makes surrogate-based analysis less tractable. Furthermore, theoretical understanding of other hyperparameters are indeed interesting topics, but we would like to bring up another significance of our work: it includes the first systematic large-scale study highlighting the problem of hyperparameter selection in data attribution, which provides important future research directions.
>
> > The experimental indicators are single, and many results are based on the synthetic indicator LDS. End-to-end verification has not been performed in actual tasks (such as noise detection and training data screening), and it cannot be proved that the selected hyperparameters can bring better downstream performance.
> > Have you considered other performance indicators? All current experiments use LDS as the only metric. Have you considered using downstream performance as an evaluation metric in data denoising or data selection tasks? If it cannot be extended to practical tasks, the practicality of this paper will be questioned.
>
> Following the reviewer’s suggestion, we include an evaluation of our proposed algorithm on downstream settings. We focus on the task of data selection, where we measure the performance drop of models when the most positively influential training data are removed (more drop indicates better attribution performance). The results are shown in the table below. We observe that our proposed algorithm generally achieves visible model performance drops, justifying the effectiveness of the algorithm in downstream tasks.
> | Settings         | Full  | R     | ID    | IS    | TD    | TS    |
> |------------------|-------|-------|-------|-------|-------|-------|
> | MNIST + LR       | 88.52 | 87.52 | 87.46 | 81.44 | 86.98 | 79.92 |
> | CIFAR2 + ResNet9 | 75.8  | 69.8  | 70.2  | 70.2  | 73.0  | 55.4  |
> Here Full means no removal is done, R means random removal, ID means IFFIM + default (regularization = 0), IS means IFFIM + selected by algorithm, TD means TRAK + default (regularization = 0), TS means TRAK + selected by algorithm. We will add more experiments on more datasets in the final version of our paper.
>
> > The proposed indicators are based on complex partial derivative conditions and high-order tensor expressions, which lack interpretability for non-theoretical researchers and have a high threshold for actual adoption.
>
> While we recognize the reviewer’s concern regarding interpretability, we have taken care to design our algorithm with a clear conceptual intuition and practical implementation. In brief, the algorithm is simply seeking a range of regularization where the surrogate indicator is close to neither of its bounds (0 and 1) when averaged over the test set. To further facilitate understanding, we also provide an intuition of the surrogate indicator here. For simplicity, we let $v$ denote the gradient of $f$ at $z’$ and $\theta^\ast_S$. By observing the definition of terms $t_{k, z’, \lambda}$ where $k = 1, 2, 3$, the surrogate indicator can be viewed as the cosine “angle” between $v$ and $(F_S+\lambda I_p)^{-1} v$. Here, $(F_S+\lambda I_p)^{-1} v$is related to the $\theta$ update direction in IFFIM, as in Equation 2. When the surrogate indicator is close to 0, this “angle” is overly large, suggesting a significant misalignment between the $\theta$ update direction and the gradient descent direction $v$. This indicates that the update direction is not reliable. In this case, increasing $\lambda$ helps intuitively because it helps condition the inverted matrix $F_S$ better. On the other hand, when the surrogate indicator is close to 1, $F_S$ is overshadowed by the regularization term $\lambda I_p$, meaning that the attribution method may become dominated by the regularization and lose sensitivity to the underlying data structure.
>
> > Are theoretical indicators generalizable? The derivation of the surrogate is based on a series of technical assumptions. Do these assumptions hold true in large-scale deep models or strong non-convex situations? It is recommended to provide further empirical support or hypothesis sensitivity analysis.
>
> We mainly focus on Assumption B.6 and Assumption B.8. We would like to state that the constants $C_1$, $C_2$, and $C_3$ generally stay at the 1e0 level instead of scaling visibly with the smallest eigenvalue. Due to computational limitations, currently we only conduct the test in the CIFAR2+ResNet9 setting, which already contains a certain level of non-convexity. We will add similar experiments and analysis on other larger settings. Note that we compute $C_1$ with respect to the smallest eigenvalue (instead of all eigenvalues), which is reasonable because only the $C_1$ term related to the smallest eigenvalue occurs in our theoretical derivation. Our results are that $C_3 = 0.4082$ is enough for all test points with $\varepsilon_g = 0.5$. Further, 90% of the test points yield $C_1 = 0.3121$ and $C_2 = 51.9933$. In comparison, the smallest eigenvalue is only about 1e-4.
>
> > The author identified the pain points of hyperparameter tuning, but the solution he finally gave is more like an "ad hoc fix" rather than a systematic solution to the problem.
>
> We would like to highlight that the surrogate indicator and the related algorithm we gave is both theoretically interpretable and practically useful. As hyperparameter selection in data attribution is different from ordinary hyperparameter selection in machine learning, our solution has a more general implication that could inspire further research in this direction, and we believe that the theoretical analysis that supports our solution could be generalized to solve similar problems in future works.
>
> > Is there a risk of "misleading selection"? The author's method attempts to fit the LDS trend with the surrogate curve, but is there a situation where the surrogate indicator fails to be monotonically in some areas, resulting in the misselection of suboptimal λ? Have the authors considered uncertainty estimates or confidence bounds to aid in the selection?
>
> We acknowledge that we do not take uncertainty into account when formulating the surrogate indicator and the algorithm, but we would like to state that the uncertainty is minimal and unlikely to affect the reliability of the selected hyperparameters in practice. To justify this, we re-run the experiment in CIFAR2+ResNet9 setting 50 times with different training seeds for the ensemble model. The mean and standard error (SE) of LDS are shown in the table below, which demonstrates that the error is generally negligible compared with the mean. Additionally, we also observe that in 49 among these 50 trials, our algorithm selects the same regularization term. We also notice that the surrogate indicator curves across all 50 trials are monotonic.
> | Regularization | 1e-1  | 1e0   | 1e1   | 1e2   | 1e3   | 1e4   |
> |----------------|-------|-------|-------|-------|-------|-------|
> | Mean           | 0.172 | 0.214 | 0.248 | 0.254 | 0.258 | 0.259 |
> | SE             | 0.003 | 0.003 | 0.003 | 0.004 | 0.004 | 0.004 |

---

> > ### Comment · Reviewer_GpMe · 2025-08-06
> >
> > Thank you for the detailed clarifications and additional experiments. I appreciate the authors’ efforts in addressing all the raised concerns. My questions have been resolved, and I will maintain my current score.

---

> > > ### Author Response · Authors · 2025-08-06
> > >
> > > Thank you for acknowledging our response! We appreciate your time and review.

---

### Official Review · Reviewer_Uabe · 2025-07-01

**Clarity:** 2
**Significance:** 3
**Originality:** 4
**Rating:** 4
**Confidence:** 4

**Summary:**

This paper identifies hyperparameter sensitivity as a critical but overlooked challenge in the research of training data attribution. The authors conduct the first large-scale empirical study demonstrating that most data attribution methods are highly sensitive to key hyperparameters like regularization terms, projection dimensions, and training epochs. The paper provides a theoretical analysis of the regularization parameter in influence function methods and proposes a practical, retraining-free selection heuristic that approximates optimal regularization values. Their experiments across multiple datasets and models show that the proposed method effectively selects good hyperparameter values without the computational cost of repeated model retraining.

**Questions:**

There is also a hyperparameter in the metric of LDS, the subset size $a$. I wonder whether varying $a$ will also significantly change the results?

**Ethical Concerns:**

["NO or VERY MINOR ethics concerns only"]

**Final Justification:**

Happy to recommend acceptance as long as the authors will incorporate the proposed changes to Section 4.1 into the revision.

**Limitations:**

It would be good to evaluate the proposed heuristic with more datasets.

**Paper Formatting Concerns:**

No concern.

**Quality:**

4

**Strengths And Weaknesses:**

**Strengths**
- This paper revisits the active research area of training data attribution and identifies a critical aspect that has been largely overlooked by the community—the computational challenge of hyperparameter tuning in data attribution methods.
- The paper is well-written and clearly structured, making it accessible to readers (with the exception of Section 4.1, discussed below).

**Weaknesses**
- Section 4.1 suffers from poor presentation and organization. The notation is confusingly introduced in separate remark paragraphs after the main theorem statement, making it difficult to parse the theoretical contributions. This section would benefit from significant revision to include: (1) clear interpretation of the theorem statement, (2) intuitive explanations of the key insights, and (3) a proof sketch for Theorem 4.3 (and other theoretical results if space allows).

- While the proposed heuristic in Section 4.2 demonstrates effectiveness in the experiments of Section 4.3, there are concerns about its generalizability. The fixed threshold of 0.5 may work well primarily because the experimental evaluation relies on relatively clean datasets (MNIST and CIFAR-10). I suspect this threshold would require significant adjustment for different types of datasets or domains. This concern is supported by the notably worse performance on the MAESTRO dataset, suggesting that the method's robustness across diverse data characteristics remains questionable.

---

> ### Author Rebuttal · Authors · 2025-07-31
>
> We appreciate the insightful comments by the reviewer.
>
> > Section 4.1 suffers from poor presentation and organization. The notation is confusingly introduced in separate remark paragraphs after the main theorem statement, making it difficult to parse the theoretical contributions. This section would benefit from significant revision to include: (1) clear interpretation of the theorem statement, (2) intuitive explanations of the key insights, and (3) a proof sketch for Theorem 4.3 (and other theoretical results if space allows).
>
> We acknowledge that Section 4.1 introduces a dense set of new notations and theoretical results within a compact space. Following the suggestion of the reviewer, we will rewrite Section 4.1 in mainly the following aspects. First, we will move the introduction of key notations (e.g. $t_{k, z’, \lambda}$) to precede Theorem 4.3, and provide intuitive explanations for each. In particular, we will interpret $\alpha_{z’, i}$ and $g_{z’}$ by relating them to the leave-one-out (LOO) interpretation of data attribution, and clarify other terms like $t$, $r$, and $o$ by expressing them as inner products of more elementary components. Then, we will adjust Theorem 4.3 and add an interpretation of the result that is grounded in the meanings of the key terms from Equation 8.Finally, we will summarize the core ideas and key proof steps in a clear proof sketch below Theorem 4.3, incorporating relevant intermediate results from the Appendix. We also consider splitting Section 4.1 into smaller subsections for better readability. These modifications will be included in the final version of our paper.
>
> > While the proposed heuristic in Section 4.2 demonstrates effectiveness in the experiments of Section 4.3, there are concerns about its generalizability. The fixed threshold of 0.5 may work well primarily because the experimental evaluation relies on relatively clean datasets (MNIST and CIFAR-10). I suspect this threshold would require significant adjustment for different types of datasets or domains. This concern is supported by the notably worse performance on the MAESTRO dataset, suggesting that the method's robustness across diverse data characteristics remains questionable.
>
> We include an additional larger experimental setting on text data (GPT-2 on WikiText-2), which validates the effectiveness of our algorithm to select a good regularization coefficient. The results are summarized in the table below.
> | Attributor/Regularization | 1e-5  | 1e-4  | 1e-3             | 1e-2  | 1e-1  | 1e0   | 1e1   |
> |---------------------------|-------|-------|------------------|-------|-------|-------|-------|
> | IFFIM                     | 0.061 | 0.081 | 0.117 (selected) | 0.145 | 0.127 | 0.085 | 0.067 |
> | LoGra                     | 0.133 | 0.158 | 0.207 (selected) | 0.253 | 0.240 | 0.170 | 0.123 |
> Here, (selected) indicates the regularization strength selected by our algorithm. The performances are measured in LDS.
>
> We highlight that our study now covers experiments on **three different modalities** (image, text, and music), as well as **three types of data attribution methods** (IFFIM, TRAK, and LoGra). These experiments support that a fixed threshold 0.5 enables our algorithm to reliably select effective regularization strengths across diverse settings.
>
> > There is also a hyperparameter in the metric of LDS, the subset size . I wonder whether varying  will also significantly change the results?
>
> We included the discussion of the subset size in LDS with empirical evidence in Appendix C.3. As a high level summary, we observed that while the magnitude of LDS could vary with the subset size, the overall monotonicity and trend as a function of $\lambda$ remains similar. This observation suggests that our method remains robust across different subset sizes.

---

> > ### Comment · Reviewer_Uabe · 2025-08-06
> >
> > Thanks the authors for the detailed response.
> >
> > Please do incorporate the proposed changes to Section 4.1 to the revision.
> >
> > An additional comment is that the title can better emphasize "attribution parameter" or "algorithm hyperparameter" to distinguish from "training hyperparameter".

---

> > > ### Author Response · Authors · 2025-08-08
> > >
> > > Thank you for your feedback and helpful suggestions. We will incorporate the proposed updates to Section 4.1 in the revision. We also appreciate your comment regarding the title and will definitely take it into consideration to better reflect our focus on hyperparameters specific to data attribution.

---

### Official Review · Reviewer_FTFU · 2025-07-02

**Clarity:** 3
**Significance:** 2
**Originality:** 3
**Rating:** 4
**Confidence:** 3

**Summary:**

This paper investigates the problem of hyperparameter sensitivity in training data attribution (TDA) methods, both from theoretical and empirical perspectives. The authors show that popular attribution methods such as Influence Functions (IF), LiSSA, TRAK, and LoGra exhibit significant sensitivity to hyperparameter choices. Moreover, the optimal hyperparameter settings are shown to be highly dependent on the specific dataset and model, and the effects of multiple hyperparameters can be entangled, i.e., the best value for one hyperparameter may depend on the value of another.

Since accurate evaluation of TDA methods typically requires multiple retraining runs, empirical tuning of hyperparameters is computationally expensive and often impractical. To address this challenge, the authors propose a theoretically grounded framework to guide hyperparameter selection. As a case study, they theoretically derive a practical algorithm for selecting the regularization parameter in the IFFIM method, e.g. an IF variant based on an approximation involving Fisher Information Matrix. Empirical results demonstrate that this algorithm can often find hyperparameter values that are close to optimal.

**Questions:**

1. Please clarify the interpretation of the terms in Equation 8 in Theorem 4.3. A more intuitive explanation would improve the accessibility of the result.
2. How do Theorem 4.3, the surrogate indicator, and Proposition 4.7 connect to the derivation of Algorithm 1? Please describe this connection more explicitly.
3. What is the computational cost of estimating the optimal hyperparameter using Algorithm 1, and how does it compare to computing the LDS metric under similar conditions?

**Ethical Concerns:**

["NO or VERY MINOR ethics concerns only"]

**Final Justification:**

Overall, the theoretical and quantitative results presented in the paper are quite compelling. The paper significantly contributes to the field by systematically evaluating the sensitivity of TDA methods. The rebuttal was helpful in clarifying the theoretical derivations.  However, my concerns about the significance of the theoretical contribution remain, as the proposed method focuses on a single parameter within one specific family of TDA methods.

**Limitations:**

The authors adequately addressed the limitations.

**Paper Formatting Concerns:**

No paper formatting concerns.

**Quality:**

3

**Strengths And Weaknesses:**

**Strengths:**

* The paper is well-written and well-structured.
* The empirical study is comprehensive, covering a variety of popular TDA methods (Influence Functions, LiSSA, TRAK, LoGra), and convincingly demonstrates hyperparameter sensitivity across datasets and models.
* The work brings valuable attention to a widespread, under-addressed issue in the TDA literature.
* The paper presents a promising strategy to mitigate this issue: using theoretical derivations to guide hyperparameter selection. The case study demonstrates how this approach can help identify a promising region in hyperparameter space without expensive evaluation metric calculations.

**Weaknesses:**

* The proposed theoretical strategy, while appealing in principle, is less convincing in the case study. The theoretical derivation involves several assumptions (e.g., the LDS being uni-modal and concave in $\lambda$, technical assumption in Appendix B.3) and heuristics (e.g., $\xi_{T, \lambda}$ in lines 263–264) that may not hold broadly in practice. The empirical results in Figure 2 show that the hyperparameter values determined by the proposed algorithm are not always close to the optimal value.
* It is unclear how easily this theory-driven approach for hyperparameter selection can be generalized to other TDA methods beyond IF.
* The derivation of Algorithm 1 lacks clarity:

  * Theorem 4.3 introduces a large number of new definitions with little intuitive explanation.
  * The purpose of Proposition 4.7 is unclear.
  * The derivations steps from the theoretical results (Theorem 4.3, Proposion 4.7) to Algorithm 1 are not clearly articulated.

**Minor Weaknesses:**

* Possible typos:

  * “The sampled subset” (line 212) — may mean “a sample of subsets.”
  * “Trained the full dataset S” (line 258)
* The finding that earlier model checkpoints may produce better results than final ones for some TDA methods (line 111) is quite interesting. I wish the authors gave possible explanations for this.

---

> ### Author Rebuttal · Authors · 2025-07-31
>
> We appreciate the positive evaluation as well as the constructive suggestions given by the reviewer.
>
> > The proposed theoretical strategy, while appealing in principle, is less convincing in the case study. The theoretical derivation involves several assumptions (e.g., the LDS being uni-modal and concave in , technical assumption in Appendix B.3) and heuristics (e.g.,  in lines 263–264) that may not hold broadly in practice. The empirical results in Figure 2 show that the hyperparameter values determined by the proposed algorithm are not always close to the optimal value.
>
> Rather than aiming to find an exact optimal regularization—which is intractable and costly in practice—we focus on providing an efficient, empirically validated method (the surrogate indicator and the algorithm) for selecting a strong candidate. The results we include demonstrate that our method outperformed the default choice (no regularization) and baselines (the spectra-based) in terms of stability and overall performance. Furthermore, our empirical studies under multiple different settings justify that the LDS curve is very likely to be uni-modal and concave in practice.
>
> > It is unclear how easily this theory-driven approach for hyperparameter selection can be generalized to other TDA methods beyond IF.
>
> As discussed in Section 2 and Appendix, many state-of-the-art data attribution methods such as TRAK and LoGra can be categorized as variants of IFFIM. The experiments in Section 4 and Appendix focus on the effectiveness of our method with IFFIM and TRAK. We also include an experiment of LoGra under GPT-2 + WikiText-2 setting, as shown in the table below. These results, obtained under multiple different settings, demonstrate the strong applicability of the method in practice.
> | Attributor/Regularization | 1e-5  | 1e-4  | 1e-3             | 1e-2  | 1e-1  | 1e0   | 1e1   |
> |---------------------------|-------|-------|------------------|-------|-------|-------|-------|
> | LoGra                     | 0.133 | 0.158 | 0.207 (selected) | 0.253 | 0.240 | 0.170 | 0.123 |
> Here, (selected) indicates the regularization strength selected by our algorithm. The performances are measured in LDS.
>
> > Theorem 4.3 introduces a large number of new definitions with little intuitive explanation.
> > Please clarify the interpretation of the terms in Equation 8 in Theorem 4.3. A more intuitive explanation would improve the accessibility of the result.
>
> To aid understanding, we now offer an intuitive explanation of both sides of Equation 8. We first focus on the surrogate indicator, which is also the RHS of Equation 8. For simplicity, we let $v$ denote the gradient of $f$ at $z’$ and $\theta^\ast_S$. By observing the definition of terms $t_{k, z’, \lambda}$ where $k = 1, 2, 3$, the surrogate indicator can be viewed as the cosine “angle” between $v$ and $(F_S+\lambda I_p)^{-1} v$. Here, $(F_S+\lambda I_p)^{-1} v$is related to the $\theta$ update direction in IFFIM, as in Equation 2. When the surrogate indicator is close to 0, this “angle” is overly large, suggesting a significant misalignment between the $\theta$ update direction and the gradient descent direction $v$. This indicates that the update direction is not reliable. In this case, increasing $\lambda$ helps intuitively because it helps condition the inverted matrix $F_S$ better. On the other hand, when the surrogate indicator is close to 1, $F_S$ is overshadowed by the regularization term $\lambda I_p$, meaning that the attribution method may become dominated by the regularization and lose sensitivity to the underlying data structure. Similarly, the LHS of Equation 8 measures the “angle” between $v$ and the ground-truth direction that involves $g_{z’}$ (by applying Lemma B.3). Hence, Theorem 4.3 is in essence stating that when the “angle” between $v$ and $(F_S+\lambda I_p)^{-1} v$ is of comparable level as the “angle” between $v$ and ground-truth direction (not too big or too small), LDS could reach its maximum. We will provide a similar and more detailed interpretation in the final version of our paper.
>
> > The purpose of Proposition 4.7 is unclear.
> > The derivation steps from the theoretical results (Theorem 4.3, Proposition 4.7) to Algorithm 1 are not clearly articulated.
> > How do Theorem 4.3, the surrogate indicator, and Proposition 4.7 connect to the derivation of Algorithm 1? Please describe this connection more explicitly.
>
> We establish a clearer connection between Theorem 4.3, Proposition 4.7 and Algorithm 1 here. Overall, Theorem 4.3 inspires the formulation of the surrogate indicator as well as Algorithm 1, while Proposition 4.7 justifies a desirable property of the surrogate indicator, which is helpful for Algorithm 1. In concrete, the surrogate indicator is introduced as exactly the RHS of Equation 8 in Theorem 4.3. The reason we drop LHS of Equation 8 is that it involves retraining-related terms $g$ and $\alpha$, whose computation is costly. From the above discussion of the interpretation of Equation 8, we notice that a good regularization strength commonly lies in the region where the surrogate indicator is close to neither of 0 and 1. Further, Proposition 4.7 guarantees that the surrogate indicator and LHS of Equation 8 both lie in the interval [0, 1] under certain assumption, which suggests that a fixed threshold of the surrogate indicator such as 0.5 could be adopted to select the regularization, which is exactly how Algorithm 1 works. Finally, Proposition 4.7 ensures that both sides of Equation 8 are comparable as they both fall in the interval [0, 1].
>
> > What is the computational cost of estimating the optimal hyperparameter using Algorithm 1, and how does it compare to computing the LDS metric under similar conditions?
>
> Notably, directly computing LDS requires retraining, making it prohibitively expensive for large-scale models. However, our method only requires gradients at test examples, which can be obtained easily through inference passes. Under CIFAR2 + ResNet9 setting, searching for a same number of hyperparameter costs 13.7s using Algorithm 1, while costing 9m12.6s when ground-truth data in LDS are computed on the test set and the hyperparameter is selected by computing LDS directly. We obtain a 40.3x speedup over LDS-based selection under the same computational condition, demonstrating the practical efficiency of our approach.
>
> > Possible typos
>
> We appreciate the reviewer’s careful reading and will ensure all identified typos are corrected in the final version.
>
> > The finding that earlier model checkpoints may produce better results than final ones for some TDA methods (line 111) is quite interesting. I wish the authors gave possible explanations for this.
>
> While a complete explanation requires thorough investigation, our hypothesis is that this phenomenon stems from numerical issues due to vanishing gradient norms in later epochs. We also include an empirical fact here that supports our hypothesis.
> | epoch              | 0     | 10    | 20    | 30    | 40    | 50    | 60    | 70    |
> |--------------------|-------|-------|-------|-------|-------|-------|-------|-------|
> | with normalization | 0.111 | 0.125 | 0.128 | 0.130 | 0.128 | 0.131 | 0.128 | 0.129 |
> | w/o normalization  | 0.104 | 0.155 | 0.115 | 0.075 | 0.075 | 0.112 | 0.100 | 0.095 |
> Here the attribution method is TracIn and normalization means that the inner product of gradients in TracIn is replaced with the cosine angle between gradients. We observe that training epoch causes the performance to drop significantly after 10 epochs in absence of normalization, implying that the issue could be due to small gradient norms.

---

> > ### Comment · Reviewer_FTFU · 2025-08-03
> >
> > Thank you for the detailed and thoughtful rebuttal. I appreciate the clarifications and additional experimental results, and I look forward to seeing them incorporated into the revised version of the paper. However, my concerns about the significance of the contribution remain, as the proposed method focuses on a single parameter within one specific family of TDA methods. Overall, I am maintaining my score at a borderline accept.

---

> > > ### Author Response · Authors · 2025-08-04
> > >
> > > Thank you for reading our response. We are glad that our clarifications and additional experiments have addressed most of your concerns.
> > >
> > > Regarding the significance, we do want to highlight that our contribution is two-fold:
> > >
> > > 1. The first large-scale study on hyperparameter sensitivity in TDA methods with novel findings: e.g., the sensitivity to training epoch, and the entanglement between projection dimension and regularization term (which also addresses an open question in TRAK). This is also the first study that points out the **unique challenge about hyperparameter selection in TDA**.
> > > 2. A practical algorithm for selecting the regularization term in influence-function-style TDA methods. We acknowledge that this is a single parameter within one family of TDA methods. However, the regularization term is one of **the most sensitive and important hyperparameters**, and the influence-function-style TDA methods are arguably **the most popular family of methods**.
> > >
> > > Thank you again for your time and review!

---

### Note · Authors · 2025-08-11

We thank the reviewers and AC for their thoughtful evaluations. This remark summarizes our contributions and key updates made during rebuttal.

Our contributions:

First, **we identify an overlooked problem: hyperparameter selection for data attribution**, which is distinct from conventional hyperparameter tuning in machine learning due to **the high computational cost associated with data attribution evaluation metrics**. We show that attribution quality is highly sensitive to these choices, warranting focused attention in this subfield.

Second, **we conduct (to our knowledge) the first comprehensive empirical study of hyperparameter sensitivity in data attribution, benchmarking widely used methods across diverse datasets and modalities**. Our study leads to several actionable findings. Notably, we answered an open question in TRAK by exploring the entanglement between the "projection dimension" and the "regularization term"; we also found “training epochs” as an overlooked yet influential hyperparameter. See more details in Sections 3.2.

Third, **we introduce a theoretically grounded algorithm that enables efficient selection of regularization strength, a common and influential hyperparameter in many SOTA data attribution methods** (Section 4). Our approach delivers clear attribution gains at markedly lower cost by avoiding the repeated retraining required by LDS-targeted tuning.

During rebuttal, we addressed reviewer concerns with new evidence and clarifications, mainly in the following aspects:
- **Expanded experiments**. (a) We added results on the larger **WikiText-2 dataset with GPT-2 model**. (b) We moved beyond LDS and **evaluated downstream utility** by adapting our approach to data selection. Both extensions indicate that the algorithm is practical and broadly applicable across settings.
- **Clearer theoretical intuition**. We improved presentation in Section 4, offering a geometric “angle” interpretation that ties terms in the equations to intuitive quantities, and reorganized the material for concision and readability.
- **Demonstrated generality**. Our method applies to multiple popular attribution techniques (TRAK, LoGra, IFFIM), spans three modalities (image, text, music), and supports different metrics (LDS and data selection).

We appreciate the reviewers’ recognition of these efforts and will incorporate all proposed updates in the camera-ready version. We hope this remark helps AC weigh the contribution and impact of our work.

---

### Decision · Program_Chairs · 2025-09-17

**Decision:**

Accept (poster)

**Comment:**

This paper investigate the issue of the sensitivity of data attribution method to its hyperparameters. The author conduct empirical studies on this issue, and not surprisingly, the sensitivity to hyper-parameters depends on the method and data. Theoretically, the author also analysed a simple special case and derive a selection procedure without model re-training. However, the concerns about the significance of this theoretical results has been raised due to its over-simplification. In addition, the clarity of the paper can be further improved. Nevertheless, this paper still provide an initial step towards this problem, and can be valuable to the community.